# CLIPTTA: Robust Contrastive Vision-Language Test-Time Adaptation

**Marc Lafon**[*,1,2]         **Gustavo A. Vargas Hakim**[*,3]         **Clément Rambour**[2]

**Christian Desrosier**[3]         **Nicolas Thome**[2,4]

[1]Conservatoire National des Arts et Métiers, CEDRIC, F-75141 Paris, France
[2]Sorbonne Université, CNRS, ISIR, F-75005 Paris, France
[3]ETS Montreal, Canada
[4]Institut universitaire de France (IUF)

## Abstract

Vision-language models (VLMs) like CLIP exhibit strong zero-shot capabilities but often fail to generalize under distribution shifts. Test-time adaptation (TTA) allows models to update at inference time without labeled data, typically via entropy minimization. However, this objective is fundamentally misaligned with the contrastive image-text training of VLMs, limiting adaptation performance and introducing failure modes such as pseudo-label drift and class collapse. We propose CLIPTTA, a new gradient-based TTA method for vision-language models that leverages a soft contrastive loss aligned with CLIP's pre-training objective. We provide a theoretical analysis of CLIPTTA 's gradients, showing how its batch-aware design mitigates the risk of collapse. We further extend CLIPTTA to the open-set setting, where both in-distribution (ID) and out-of-distribution (OOD) samples are encountered, using an Outlier Contrastive Exposure (OCE) loss to improve OOD detection. Evaluated on 75 datasets spanning diverse distribution shifts, CLIPTTA consistently outperforms entropy-based objectives and is highly competitive with state-of-the-art TTA methods, outperforming them on a large number of datasets and exhibiting more stable performance across diverse shifts. Source code is available at: CLIPTTA Repository.

## 1   Introduction

Vision-language models (VLMs), such as CLIP [1] and ALIGN [2], are multimodal foundation models with strong zero-shot performance in downstream classification tasks. Yet, their ability to generalize to specialized domains, *e.g*., medical imaging or corrupted inputs, remains limited without adaptation, making this an active area of research.

Test-Time Adaptation (TTA) addresses the adaptation of pre-trained models to new downstream tasks during inference, without access to ground-truth labels, typically by updating model parameters via gradient-based optimization [3, 4, 5, 6, 7]. This label-free adaptation is particularly valuable for deploying VLMs in real-world applications where annotation is scarce and costly, such as medical image processing [8], human-robot interaction [9], and federated learning [10].

Entropy minimization is the most common TTA objective [11, 12, 13, 14], as it mirrors the cross-entropy training of standard classifiers. However, it is fundamentally misaligned with the contrastive

---

[*] Equal contribution
 Corresponding author: marc.lafon@lecnam.net

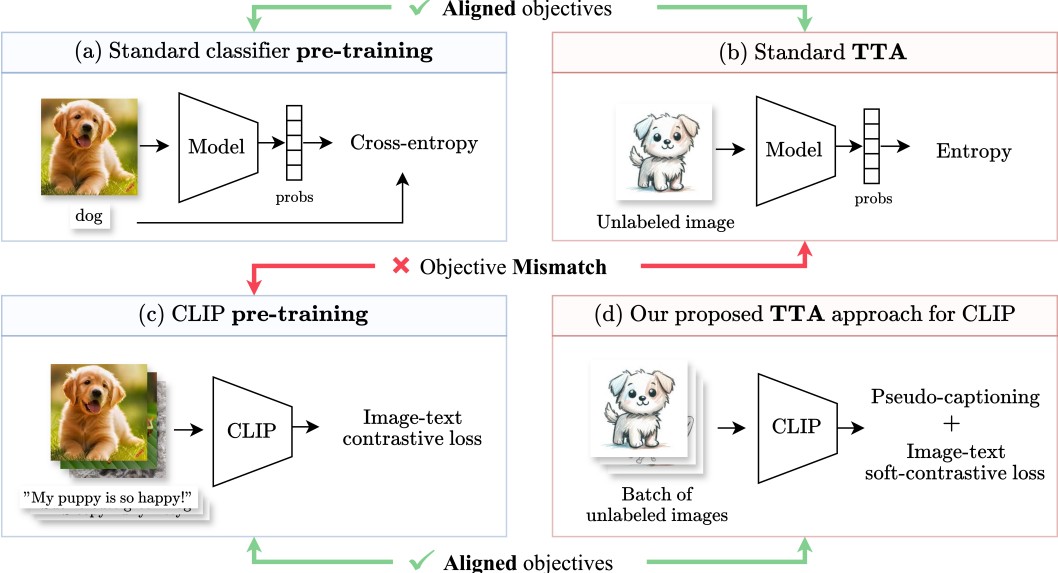

Figure 1: **Motivation for CLIPTTA**. Standard test-time adaptation (TTA) methods often rely on entropy minimization (b), which aligns with the cross-entropy loss used in classifier training (a) but is misaligned with CLIP's contrastive pre-training (c), hindering adaptation due to incompatible gradient dynamics. CLIPTTA instead uses a soft contrastive loss aligned with CLIP's objective, reinforcing alignment between images and their predicted pseudo-captions within the batch (d). Our gradient analysis shows that this contrastive, batch-aware formulation improves robustness to pseudo-label drift and class collapse—two failure modes common to entropy-based TTA methods.

image-text pre-training objective of VLMs like CLIP, as illustrated in Fig. 1, potentially hindering adaptation due to differing gradient dynamics. Recent works have attempted to improve TTA of CLIP by leveraging visual-textual similarities in a transductive manner [6, 7], yet the objective misalignment remains unresolved. Furthermore, when labeled data is available, recent work on fine-tuning [15] demonstrates that using the exact same loss function as during CLIP pre-training leads to better performance on downstream tasks.

This mismatch in objectives leads to fundamental issues during adaptation: entropy-minimization methods are prone to *pseudo-label drift*, where the model reinforces its own mistakes. This can lead to *class collapse*, where predictions concentrate on a narrow set of classes regardless of the input [16, 13], severely hindering adaptation. Numerous efforts have been made to reduce the adverse impact of pseudo-label misclassification [12, 13, 14, 3]. However, these methods make predictions for each sample independently, without accounting for other predictions in the batch, which limits their robustness. This becomes especially critical when the source model's accuracy is low or when input batches contain out-of-distribution (OOD) samples that belong to unknown classes [17, 12, 18].

Together, these observations raise a central question:

> *How to design an adaptation loss that is more suited for gradient-based test-time adaptation of contrastive vision-language models such as CLIP?*

In this work, we introduce CLIPTTA, a new test-time adaptation method tailored to vision-language models. It employs a soft contrastive image-text loss that mirrors CLIP's pre-training objective, providing natural continuity in adaptation. As illustrated in Fig. 1, this design reflects our central assumption: adaptation losses should align with the model's multimodal contrastive training paradigm. Importantly, the contrastive nature of the CLIPTTA loss links predictions within a batch, incorporating mechanisms to mitigate the risk of class collapse caused by noisy pseudo-labels. It also demonstrates increased robustness in open-set scenarios, where both in-distribution (ID) and out-of-distribution (OOD) samples are present. We further augment it with a discriminative loss to separate ID from OOD samples, improving performance under open-set conditions.

Our contributions can be summarized as follows:

- We introduce CLIPTTA, a new TTA method for CLIP based on a soft contrastive image-text loss aligned with its pre-training objective, offering a principled alternative to entropy minimization.
- We provide a theoretical analysis of CLIPTTA's gradients, showing how its batch-aware design improves robustness to pseudo-label drift and class collapse—two key failure modes of standard gradient-based TTA methods.
- We extend CLIPTTA to open-set adaptation with an Outlier Contrastive Exposure (OCE) loss, improving ID/OOD separation and robustness under distribution shift.

We conduct extensive benchmarking across 75 diverse datasets, spanning four types of distribution shifts: corruptions, domain shifts, coarse-grained, and fine-grained classification. Empirical results show that our soft contrastive loss consistently outperforms entropy-based objectives for gradient-based TTA of vision-language models, establishing it as a more effective alternative. In addition, CLIPTTA is highly competitive with state-of-the-art TTA methods, outperforming them on a large number of datasets and exhibiting more stable performance across diverse shifts. It also achieves notable gains in accuracy and OOD detection under open-set conditions.

## 2 Related work

**Test-time adaptation** (TTA) seeks to adapt a model to new datasets *on the fly* in the absence of labels. This process is performed on independent data streams that showcase only a small portion of the full data distribution. Aiming to adapt deep classifiers to new domains, TENT [11] proposed the widely exploited technique of entropy minimization. The entropy loss is chosen for its link with cross-entropy, with the intent of extending the model's training in an unsupervised way. Building on this principle, several approaches have been proposed: filtering out unimportant samples based on an entropy criterion in ETA [12], and further filtering those with small gradients in SAR [13], minimizing the marginal distribution's entropy across image transformations in MEMO [19], meta-learning the TENT loss via conjugate pseudo-labels [20], storing the most confident samples in memory for a *cleaner* adaptation in RoTTA [14], or combining entropy minimization with a clustering loss constraint in TTC [21]. While these methods rely on additional mechanisms such as filtering or confidence-based selection, CLIPTTA achieves robustness to pseudo-label drift and collapse by a simple modification of the adaptation objective. Contrastive learning approaches have also been explored, such as AdaContrast [22], where a student-teacher model is trained using pseudo-labels obtained from weak and strong image augmentations as in MoCo [23]. In contrast, our contrastive adaptation refers to visual-text interactions in the context of VLMs. To the best of our knowledge, this is the first attempt to explore this particular contrastive TTA formulation for VLMs.

**TTA for VLMs.** Several methods have been proposed to adapt VLMs to new streams of unseen data. CLIPArTT [6] introduces a new loss function specifically tailored to VLMs, combining image-to-image and text-to-text similarities to generate pseudo-labels and utilizing a small subset of probable classes to form new image-wise text prompts. WATT [7] extends this idea with prompt ensembling and weight averaging. While CLIPArTT's loss better leverages CLIP's multimodal structure than entropy minimization, it remains heuristically driven and loosely aligned with CLIP's contrastive training objective. Complementary to these, other methods explore alternative adaptation paradigms. TPT [3] performs adaptation through prompt tuning [24]: rather than updating the model's internal weights, it optimizes a small set of text prompts using entropy minimization. Although it uses gradient-based adaptation, this approach is fundamentally distinct from traditional TTA methods that typically update normalization parameters, and it comes with a high computational cost due to its reliance on multiple augmentations per image. TDA [4] adopts an even more distinct approach: it operates in a gradient-free manner by building positive and negative caches of past predictions, which are then used as pseudo-labels to simulate few-shot episodes as in [25]. While TDA achieves strong results on Imagenet variants, we found it to perform poorly under other types of distribution shifts, such as corruptions. In contrast, our approach, CLIPTTA, requires only a simple modification of the loss function and delivers robust performance across all TTA benchmarks.

**Open-set TTA** is a more challenging branch of TTA, where batches are polluted with out-of-distribution (OOD) samples that belong to unknown classes. Open-set TTA methods aim at detecting

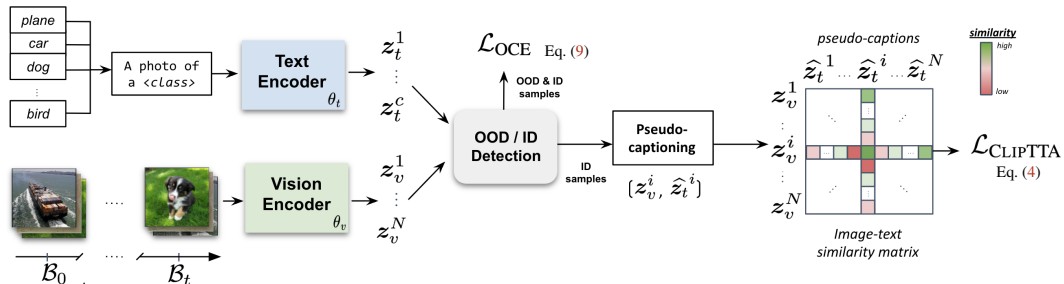

Figure 2: **Illustration of CLIPTTA**. CLIPTTA in Sec. 3.1 consists of a soft contrastive loss specifically designed for TTA of VLMs like CLIP. We show in Sec. 3.2 that CLIPTTA is robust to class collapse and pseudo-label errors. Finally, we add an OCE loss to be robust to OOD samples in batches in Sec. 3.3 and improve the ID/OOD detection and accuracy in open-set scenarios.

OOD samples from in-distribution (ID) samples, and improve the model's accuracy on ID images. OSTTA [17] uses an entropy heuristic based on a student-teacher model to disregard OOD samples and apply entropy minimization on the ID ones. SoTTA [26] uses the maximum predicted probability to filter and store the most confident samples in memory, and applies TENT on them. On the contrary, STAMP [27] filters samples and their augmentations via entropy, to also preserve them in a memory for entropy minimization. UniEnt [28] addresses the problem more explicitly by modeling the samples' outlier score as a mixture of two Gaussian distributions, later using entropy minimization on the ID samples and entropy maximization on the OOD samples. As in the closed-set scenario, these methods do not transfer optimally to VLMs, since entropy does not connect well with CLIP's pre-training loss. Our adaptation loss aligns with CLIP's pre-training, and we propose a discriminative OOD loss that directly aligns with ID/OOD separations metrics.

## 3 CLIPTTA

We introduce CLIPTTA, a contrastive test-time adaptation method tailored to VLMs such as CLIP, as illustrated in Fig. 2. By aligning the adaptation objective with CLIP's image-text contrastive pre-training described in Sec. 3.1, CLIPTTA improves robustness to pseudo-label errors and class collapse through its batch-aware formulation, as demonstrated by our gradient analysis in Sec. 3.2. Combined with the Outlier Contrastive Exposure loss introduced in Sec. 3.3, it improves both OOD detection and accuracy for robust adaptation in open-set scenarios.

### 3.1 Contrastive adaptation loss at test-time

Let us denote CLIP's visual encoder as $f_{\theta_v}^v(\cdot)$ and its textual encoder as $f_{\theta_t}^t(\cdot)$, with model parameters $\theta = (\theta_v, \theta_t)$. Given an image $\boldsymbol{x}$ and a textual prompt $\boldsymbol{t}$, the normalized visual and text features are $\boldsymbol{z}_v = f_{\theta_v}^v(\boldsymbol{x})$ and $\boldsymbol{z}_t = f_{\theta_t}^t(\boldsymbol{t})$. To classify an image in a downstream task, we construct class-specific captions of the form $\boldsymbol{t}_c = $ "A photo of a $< class >$" for each class $c$, and compute the probability of classifying image $\boldsymbol{x}_i$ as class $c$:

$$q(\boldsymbol{t}_c|\boldsymbol{x}_i) = \frac{\exp(\boldsymbol{z}_v^{i\top}\boldsymbol{z}_t^c/\tau)}{\sum_{k=1}^{C}\exp(\boldsymbol{z}_v^{i\top}\boldsymbol{z}_t^k/\tau)}, \tag{1}$$

where $\tau$ is a fixed temperature parameter.

Since ground truth captions are unavailable at test-time, we generate pseudo-captions for a batch of $N$ samples $\{\boldsymbol{x}_i\}_{i=1}^N$ by associating each image $\boldsymbol{x}_i$ to the caption of its predicted class $\hat{\boldsymbol{t}}_i = \boldsymbol{t}_{\hat{c}}$, where $\hat{c} = \arg\max_c q(\boldsymbol{t}_c|\boldsymbol{x}_i)$. We denote $\hat{\boldsymbol{z}}_t^i$ the representation of $\hat{\boldsymbol{t}}_i$. Given two pseudo-labeled image-text pairs $(\boldsymbol{x}_i, \hat{\boldsymbol{t}}_i)$ and $(\boldsymbol{x}_j, \hat{\boldsymbol{t}}_j)$, we define $p(\hat{\boldsymbol{t}}_j|\boldsymbol{x}_i)$ and $p(\boldsymbol{x}_j|\hat{\boldsymbol{t}}_i)$ as the probabilities that $\boldsymbol{x}_i$ matches $\hat{\boldsymbol{t}}_j$ and that $\hat{\boldsymbol{t}}_i$ matches $\boldsymbol{x}_j$, respectively:

$$p(\hat{\boldsymbol{t}}_j|\boldsymbol{x}_i) = \frac{\exp(\boldsymbol{z}_v^{i\top}\hat{\boldsymbol{z}}_t^j/\tau)}{\sum_{l=1}^{N}\exp(\boldsymbol{z}_v^{i\top}\hat{\boldsymbol{z}}_t^l/\tau)} \quad \text{and} \quad p(\boldsymbol{x}_j|\hat{\boldsymbol{t}}_i) = \frac{\exp(\boldsymbol{z}_v^{j\top}\hat{\boldsymbol{z}}_t^i/\tau)}{\sum_{l=1}^{N}\exp(\boldsymbol{z}_v^{l\top}\hat{\boldsymbol{z}}_t^i/\tau)}. \tag{2}$$

Although Eq. (1) and Eq. (2) appear similar, *they differ in their softmax normalization*: Eq. (1) normalizes over $C$ classes, while Eq. (2) normalizes over the $N$ predicted classes in the batch.

A natural strategy for adapting CLIP at test time is to reuse its contrastive loss on pseudo-labeled image-text pairs $(\boldsymbol{x}_i, \hat{\boldsymbol{t}}_i)$. However, this assumes pseudo-labels are correct and ignores uncertainty in the predictions. Instead, we retain alignment with CLIP's training objective while relaxing reliance on hard pseudo-labels. To this end, we introduce a soft contrastive loss that leverages the full distribution over pseudo-captions:

$$\mathcal{L}_{\text{s-cont}}(\theta) := \sum_{i=1}^{N} \Big[ \underbrace{- \sum_{j=1}^{N} p(\hat{\boldsymbol{t}}_j|\boldsymbol{x}_i) \log p(\hat{\boldsymbol{t}}_j|\boldsymbol{x}_i)}_{\text{image}\rightarrow\text{text}} \quad \underbrace{- \sum_{j=1}^{N} p(\boldsymbol{x}_j|\hat{\boldsymbol{t}}_i) \log p(\boldsymbol{x}_j|\hat{\boldsymbol{t}}_i)}_{\text{text}\rightarrow\text{image}} \Big]. \tag{3}$$

This loss retains CLIP's contrastive structure while explicitly modeling uncertainty in pseudo-labels. As shown in Fig. 2, the first term computes the entropy over the image-to-text probability distribution (row-wise), and the second term the entropy over the text-to-image probability distribution (column-wise) within the batch. Analogous to entropy minimization, which replaces hard cross-entropy with a soft and uncertainty-aware loss, our soft contrastive loss is a principled extension of the VLMs' contrastive scheme. Furthermore, it demonstrates enhanced robustness to pseudo-label errors, as studied in Sec. 3.2. To ensure fair comparisons, we use only the image-to-text term of Eq. (3) in the main experiments, as most gradient-based TTA methods update only the visual encoder. The effect of simultaneously updating the text encoder is evaluated in Appendix C.

**Final training objective.** Following prior TTA research [29, 17, 27, 14], we also incorporate standard techniques such as entropy regularization and a class-wise confident memory (CCM) to enhance adaptation. The regularization loss, based on negative marginal entropy, diversifies the predictions by uniformizing the prediction distribution across classes. Defining $\bar{q}_c = \frac{1}{N} \sum_{i=1}^{N} q(\boldsymbol{t}_c|\boldsymbol{x}_i)$ as the batch-wise average probability for class $c$ (*i.e.*, over probabilities in Eq. (1)), the regularization loss is $\mathcal{L}_{\text{reg}}(\theta) = \sum_{c=1}^{C} \bar{q}_c \log \bar{q}_c$. The final CLIPTTA loss integrates the soft-contrastive loss Eq. (3), the regularization term, and the CCM memory. Memory batches $\mathcal{M}$, equal in size to test batches, are used to compute the adaptation loss:

$$\mathcal{L}_{\text{CLIPTTA}}(\theta) = \frac{1}{2} \Big[ \mathcal{L}_{\text{s-cont}}(\theta) + \mathcal{L}_{\text{s-cont}}^{\mathcal{M}}(\theta) \Big] + \lambda_{\text{reg}} \mathcal{L}_{\text{reg}}(\theta), \tag{4}$$

where $\mathcal{L}_{\text{s-cont}}^{\mathcal{M}}(\theta)$ is the soft-contrastive loss computed on the memory batch, and $\lambda_{\text{reg}}$ controls the regularizer's strength. By averaging the loss over current and memory batches, the method effectively leverages confident past predictions to improve adaptation while reducing sensitivity to noisy data.

## 3.2 Gradient Analysis

We analyze the gradient of the soft contrastive loss $\mathcal{L}_{\text{s-cont}}$ to understand how it enables robust test-time adaptation, particularly in the presence of pseudo-label errors and class imbalance. The key insight is that, unlike entropy-based losses, $\mathcal{L}_{\text{s-cont}}$ is batch-aware, allowing the model to dynamically correct prediction errors and reducing the risk of class collapse.

**Proposition 3.1** (Gradient of Soft-Contrastive Loss)**.** *Let $N_k$ be the number of samples in the batch pseudo-labeled as class $k$, and $q_{ik} = q(\boldsymbol{t}_k|\boldsymbol{x}_i)$ as in Eq. (1). The gradient of $\mathcal{L}_{\text{s-cont}}$ w.r.t. $\boldsymbol{z}_v^i$ is:*

$$\nabla_{\boldsymbol{z}_v^i} \mathcal{L}_{\text{s-cont}} = \sum_{j=1}^{N} \beta_{i,j} [-\widehat{\boldsymbol{z}}_t^j + \sum_{k=1}^{C} w_{k,i}\, \boldsymbol{z}_t^k], \tag{5}$$

$$\text{where} \quad \beta_{i,j} = p(\hat{\boldsymbol{t}}_j|\boldsymbol{x}_i)[1 + \log p(\hat{\boldsymbol{t}}_j|\boldsymbol{x}_i)], \quad \text{and} \quad w_{k,i} = \frac{N_k\, q_{ik}}{\sum_{c=1}^{C} N_c\, q_{ic}}.$$

*Proof.* See Appendix A.1. □

This expression shows that the gradient for sample $x_i$ aggregates contributions from all pseudo-captions in the batch, each weighted by $\beta_{i,j}$. Each contribution consists of two effects. The first term, $-\hat{z}_t^j$, acts as an attractive force pulling $z_v^i$ toward pseudo-caption $\hat{t}_j$. In contrast, the second term, $\sum_k w_{k,i} z_t^k$, introduces a repulsive force that pushes the embedding away from dominant class directions since classes that are more frequently predicted exert stronger repulsion.

Importantly, the coefficients $\beta_{i,j}$ are key to allowing the gradient of a sample to point toward a class different from its pseudo-label, enabling error correction by leveraging predictions from other samples in the batch, as illustrated on a toy dataset in Appendix B. They amplify the contribution of confident and semantically similar pairs in the gradient update, allowing the model to rely on more reliable examples. For instance, if $x_i$ is misclassified as class $k'$ but is close to another sample $x_j$ whose pseudo-caption reflects the correct class $k$, a large $\beta_{i,j}$ steers the update toward $z_t^k$. Such correction is not achievable by sample-wise objectives like TENT, which systematically reinforce the predicted class regardless of its correctness.

**Proposition 3.2** (Gradient Vanishing under Class Imbalance). *As one class k dominates the batch $(N_k \to N)$, the gradient of $\mathcal{L}_{s\text{-}cont}$ vanishes:*

$$||\nabla_{z_v^i} \mathcal{L}_{s\text{-}cont}|| \underset{N_k \to N}{\to} 0. \tag{6}$$

*Proof.* See Appendix A.1. $\square$

To further illustrate, consider a binary classification setting with classes $a$ and $b$, where $a$ is the most predicted class in the batch (i.e., $N_a \gg N_b$). In that case, the gradient in Eq. (5) becomes:

$$\nabla_{z_v^i} \mathcal{L}_{s\text{-}cont} = [\beta_{i,a} q_{ib} - \beta_{i,b} q_{ia}] \frac{N_a N_b}{N_a q_{ia} + N_b q_{ib}} (z_t^b - z_t^a). \tag{7}$$

The magnitude of this gradient depends on batch composition. As class imbalance grows, the coefficient $\frac{N_a N_b}{N_a q_{ia} + N_b q_{ib}}$ becomes smaller, reducing the overall gradient magnitude.

This self-regulation property acts as a built-in dampening mechanism that slows adaptation before collapse occurs, helping prevent convergence to dominant classes, preserving stable updates, and giving the model a chance to recover from poor pseudo-labeling. In contrast, entropy-based objectives such as TENT continue to reinforce dominant class predictions even as imbalance increases, accelerating collapse rather than preventing it (see derivation in Appendix A.1).

### 3.3 Outlier Contrastive Exposure loss

In this section, we extend CLIPTTA to the open-set setting, where the model is exposed to batches composed of images from both *known* classes (ID samples) and *unknown* classes (OOD samples) during adaptation. Our primary objective is to design an effective ID/OOD filtering mechanism to focus adaptation on ID samples only. For that purpose, we use the MCM [30] score, defined as $s_i = \max_c q(t_c|x_i)$, which is the most popular OOD scoring function in the context of OOD detection for VLMs. For an input image $x_i$, we further define the *outlierness* filtering weight:

$$w_i = \text{sigmoid}(s_i - \alpha), \tag{8}$$

where $\alpha$ is an adaptive and learnable threshold. Using these weights, an image $x_i$ will be considered reliable if $w_i > 0.5$ and will be regarded as OOD otherwise.

While effective OOD filtering helps to improve TTA performance in an open-set setting, we argue that we can leverage filtered-out OOD samples to improve the ID/OOD detection performance during adaptation. To this end, we introduce the Outlier Contrastive Exposure (OCE) loss that aims at improving the OOD score separation between ID and OOD samples:

$$\mathcal{L}_{\text{OCE}} = -\left[ \underbrace{\frac{\sum_{i=1}^{N} w_i s_i}{\sum_{i=1}^{N} w_i}}_{\mu_{\text{id}}} - \underbrace{\frac{\sum_{i=1}^{N}(1 - w_i)s_i}{\sum_{i=1}^{N}(1 - w_i)}}_{\mu_{\text{ood}}} \right]^2. \tag{9}$$

In the open-set scenario, our optimization objective then becomes $\min_{\theta,\alpha} \mathcal{L}_{\text{CLIPTTA}} + \lambda_{\text{oce}} \mathcal{L}_{\text{OCE}}$, where we update the parameters of the model $\theta$ and the ID / OOD threshold parameter $\alpha$ in an end-to-end fashion. Our OCE loss differs from the UniEnt loss [28] since it is purely discriminative, enforcing a more direct separation between ID and OOD features, and since it learns the separation threshold $\alpha$.

## 4 Experiments

**Datasets.** CLIPTTA is evaluated on four families of adaptation benchmarks: corruptions (CIFAR-10/100-C, Imagenet-C) with 15 perturbations, domain shifts (VisDA-C, PACS, OfficeHome, Imagenet-Domains), semantic datasets, including coarse- (CIFAR-10/100) and fine-grained classification (Imagenet, and 10 datasets from the CLIP zero-shot suite). In total, this represents a thorough evaluation over 75 datasets. A detailed description is provided in Appendix C.2. In open-set TTA, SVHN and Places-365 serve as OOD counterparts for CIFAR-10/100 and Imagenet, respectively.

**Metrics.** We report classification accuracy as the primary performance metric. In the open-set setting, we additionally report the area under the ROC curve (AUC) and the false positive rate at a 95% of ID true positive rate (FPR95) as OOD detection metrics.

**Implementation details.** We use ViT-B/16 as CLIP's backbone in all experiments. Adaptation is performed with batches of 128 images using the Adam optimizer and a learning rate of $10^{-4}$ over 10 iterations. Experiments are conducted in a non-episodic manner, *i.e.*, without restoring the model's parameters after each batch. Following the standard TTA protocol, we adapt the affine parameters of the visual encoder's normalization layers. In the open-set setting, we add 128 OOD images per batch, as done in prior work [28, 27]. The regularization and OCE losses' weights are set to $\lambda_{reg} = 1$ and $\lambda_{oce} = 1$, respectively. We validate that CLIPTTA is stable to variations of its hyper-parameters in Appendix C. Experiments were performed on two NVIDIA V100 32GB GPUs.

### 4.1 Main results

**What is the best loss function for TTA of CLIP?** We compare CLIPTTA with the two prevailing families of test-time objectives: (i) TENT-style losses, including TENT [11], ETA [12], SAR [13] and RoTTA [14], and (ii) CLIPArTT-derived losses, such as CLIPArTT [6] and WATT [7]. Table 1

| | Corruptions | | | | Domain shifts | | | | |
|---|---|---|---|---|---|---|---|---|---|
| | C-10-C | C-100-C | Imagenet-C | Average | VisDA-C | PACS | OfficeHome | Imagenet-D | Average |
| CLIP [1] | 60.2 | 35.2 | 25.5 | 40.3 | 87.1 | 96.1 | 82.5 | 59.4 | 81.3 |
| TENT [11] | 56.4 | 31.4 | 17.6 | 35.1 | 89.3 | 96.6 | 83.4 | 60.2 | 82.3 |
| ETA [12] | 61.3 | 38.9 | 26.8 | 42.3 | 88.3 | 96.7 | 84.1 | 59.9 | 82.3 |
| SAR [13] | 67.8 | 43.2 | 33.6 | 48.2 | 87.8 | 96.2 | 83.8 | 60.6 | 82.1 |
| RoTTA [14] | 58.0 | 33.6 | 24.6 | 38.7 | 83.7 | 95.8 | 82.5 | 61.6 | 80.9 |
| CLIPArTT [6] | 68.1 | 48.0 | 33.3 | 49.8 | 84.1 | 96.3 | 82.0 | 60.7 | 80.8 |
| WATT [7] | 66.0 | 38.5 | 26.0 | 43.5 | 87.7 | 96.2 | 83.4 | 61.8 | 82.1 |
| CLIPTTA (ours) | **80.7** | **52.6** | **41.1** | **58.1** | **89.6** | **97.5** | **84.2** | **63.4** | **83.7** |

Table 1: **Comparison with gradient-based TTA methods**. CLIPTTA outperforms entropy minimization methods [11, 12, 13, 14] and CLIP-specific TTA methods based on CLIPArTT's loss [6, 7] on all corruptions and domain shift datasets.

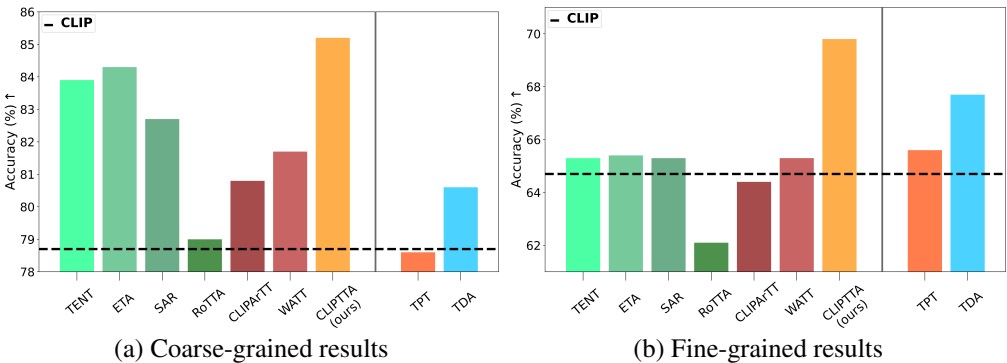

(a) Coarse-grained results      (b) Fine-grained results

Figure 3: **TTA results on semantic datasets**. Top-1 accuracy on coarse-grained datasets (CIFAR-10 and CIFAR-100) (a) and on 11 fine-grained datasets (including Imagenet) (b). Comparison with gradient-based TTA methods [11, 12, 13, 14, 6, 7] and alternative state-of-the-art TTA methods [3, 4].

reports top-1 accuracy under synthetic corruptions and real domain shifts. We highlight two key findings. First, CLIPTTA substantially improves over zero-shot CLIP, with large gains when initial accuracy is low: +20.5 pts on CIFAR-10-C, +17.4 pts on CIFAR-100-C, and +15.6 pts on Imagenet-C. In contrast, other methods perform poorly under the same conditions, which can be attributed to the increased likelihood of class collapse and pseudo-label drift when initial accuracy is low. Further analysis and finer-grained experimental evidence are presented in Appendix A.1 and Appendix B, respectively. Second, CLIPTTA is the only method that consistently achieves top performance across all benchmarks. While competing methods demonstrate strengths in specific scenarios, they fall short overall. For example, ETA performs best among the TENT-style methods on domain-shift datasets but still lags by 1.4 pts on average. Similarly, CLIPArTT is most competitive on corruption benchmarks but remains 8.3 pts behind. This trend persists across both coarse- and fine-grained datasets (Fig. 3, with extended results in Appendix C.3). Altogether, these results establish our soft contrastive loss as the most reliable and broadly effective objective for gradient-based TTA of CLIP.

**How does CLIPTTA perform against other CLIP-based TTA methods?** We further benchmark CLIPTTA against two recent state-of-the-art TTA methods tailored to CLIP: TPT [3], which adapts through text prompt tuning instead of updating normalization parameters, and TDA [4], a gradient-free approach that adjusts CLIP's logits using cached predictions. As shown in Table 2, CLIPTTA improves top-1 accuracy by an average of +19.4 pts over TPT and +15.6 pts over TDA. It achieves state-of-the-art results on nearly all domain-shift benchmarks, with the sole exception of the ImageNet-D suite, where TDA benefits from per-dataset hyperparameter tuning. However, we note that TDA lags far behind on corruption datasets, a standard benchmark in TTA, with an average gap of over 15 pts compared to CLIPTTA. Figure 3 further confirms CLIPTTA's advantage across both coarse- and fine-grained recognition tasks. While TPT and TDA are designed for single-image batches, our analysis in Appendix C.3 shows that CLIPTTA remains competitive even in this challenging setting, thanks to its use of the CCM memory and maintains stable performance across a wide range of batch sizes.

| | Corruptions | | | | Domain shifts | | | | |
|---|---|---|---|---|---|---|---|---|---|
| | C-10-C | C-100-C | Imagenet-C | Average | VisDA-C | PACS | OfficeHome | Imagenet-D | Average |
| CLIP [1] | 60.2 | 35.2 | 25.5 | 40.3 | 87.1 | 96.1 | 82.5 | 59.4 | 81.3 |
| TPT [3] | 58.0 | 33.6 | 24.6 | 38.7 | 85.0 | 94.0 | 81.7 | 62.4 | 80.8 |
| TDA [4] | 63.4 | 37.4 | 26.8 | 42.5 | 86.6 | 96.1 | 83.0 | **65.0** | 82.8 |
| CLIPTTA (ours) | **80.7** | **52.6** | **41.1** | **58.1** | **89.6** | **97.5** | **84.2** | 63.4 | **83.7** |

Table 2: **Comparison with other CLIP-based TTA methods**. CLIPTTA outperforms TPT and TDA on most corruptions and domain shifts datasets and is second best on Imagenet-D.

**How does CLIPTTA perform in the presence of semantic OOD samples?** Table 3 presents results on the open-set scenario on Imagenet, where OOD detection needs to be performed alongside classification. First, we note that all closed-set methods, except ours, perform noticeably worse than zero-shot CLIP in OOD detection, highlighting CLIPTTA's strong robustness to OOD sample contamination during adaptation. Notably, TENT and CLIPArTT suffer severe performance degradation in both classification and OOD detection, likely due to outlier interference in their pseudo-labeling process. Second, when equipped with our OCE loss, CLIPTTA consistently outperforms specialized open-set TTA methods, which use heuristic OOD

|  | ACC↑ | AUC↑ | FPR95↓ |
|---|---|---|---|
| CLIP [1] | 66.7 | 90.1 | 43.8 |
| TENT [11] | 12.4 | 49.9 | 89.4 |
| ETA [12] | 67.1 | 89.6 | 46.1 |
| SAR [13] | 58.8 | 62.0 | 75.7 |
| CLIPArTT [6] | 31.2 | 61.1 | 87.5 |
| WATT [7] | 67.1 | 87.4 | 53.4 |
| TDA [4] | 66.8 | 82.1 | 59.8 |
| CLIPTTA (ours) | **67.6** | 93.5 | 25.7 |
| OSTTA [17] † | 66.9 | 84.9 | 59.2 |
| SoTTA[26]† | 66.7 | 89.3 | 47.1 |
| STAMP [27] † | 29.7 | 63.0 | 80.2 |
| UniEnt [28] † | 65.2 | 95.4 | 17.1 |
| CLIPTTA + OCE (ours) † | **67.6** | **97.7** | **9.7** |

Table 3: Open-set TTA results with Imagenet as ID dataset and Places as OOD dataset. † denotes open-set TTA methods.

detection mechanisms, achieving +2.3 points AUC over UniEnt and +8.4 points AUC over SoTTA. Our soft contrastive objective reliably preserves and improves both accuracy and OOD detection, unlike these entropy-based methods, which tend to degrade CLIP's initial performance. Additional results on other datasets are reported in Appendix C.3.

## 4.2 Model analysis

**Ablation study.** Table 4 presents an ablation of CLIPTTA 's components on four closed-set benchmarks. The first key observation is that the soft contrastive loss ($\mathcal{L}_{\text{s-cont}}$) *alone* accounts for the vast majority of the overall performance gains, demonstrating the importance of aligning the adaptation objective with CLIP's pre-training. In low-accuracy settings, $\mathcal{L}_{\text{s-cont}}$ significantly outperforms TENT, achieving gains of +19.4 points on CIFAR-100-C and +22.7 points on ImageNet-C, where

|  | C-100 | C-100-C | IN | IN-C | FG | Avg. |
|---|---|---|---|---|---|---|
| CLIP | 68.1 | 35.2 | 66.7 | 25.5 | 64.8 | 52.1 |
| TENT | 72.9 | 31.4 | 66.5 | 17.6 | 65.1 | 50.7 |
| $\mathcal{L}_{\text{s-cont}}$ | 74.2 | 50.8 | 68.8 | 40.3 | 68.3 | 60.5 |
| $\mathcal{L}_{\text{s-cont}} + \mathcal{L}_{\text{reg}}$ | 74.9 | 52.4 | 69.1 | 38.6 | 69.8 | 60.9 |
| $\mathcal{L}_{\text{s-cont}} + \mathcal{L}_{\text{reg}} + \mathcal{M}$ | **75.3** | **52.6** | **69.6** | **41.1** | **69.9** | **61.7** |

Table 4: **Ablation analysis.** Accuracy in the closed-set setting on CIFAR-100, CIFAR-100-C, Imagenet, Imagenet-C, and 11 fine-grained datasets (FG).

TENT even degrades CLIP's performance. This confirms the vulnerability of entropy-based methods to pseudo-label drift and class collapse and supports the enhanced robustness of $\mathcal{L}_{\text{s-cont}}$, as theoretically analyzed in Sec. 3.2. On average, $\mathcal{L}_{\text{s-cont}}$ improves over TENT by +9.8 pts. Adding the regularization loss ($\mathcal{L}_{\text{reg}}$) further improves overall results by +0.4 pts, while incorporating the confident memory ($\mathcal{M}$) brings additional +0.8 pts gains.

**On CLIPTTA's robustness.** Figure 4 provides empirical insights supporting the gradient analysis in Sec. 3.2, by illustrating CLIPTTA's stability and robustness over batches on CIFAR-10-C. CLIPTTA is the only method that steadily improves accuracy throughout adaptation while all competing objectives plateau or degrade (Fig.4a). This stability is closely linked to CLIPTTA's ability to maintain prediction diversity. As shown in Fig.4b, prediction entropy remains nearly constant for CLIPTTA, whereas TENT exhibits a sharp drop in entropy, indicating collapse toward a small subset of classes. This behavior results in harmful label drift. Fig. 4c tracks the deterioration ratio, defined as the fraction of initially correct predictions that become incorrect during adaptation. TENT reaches over 30% deterioration, compared to less than 7% with CLIPTTA. These findings confirm the significant stabilizing effect of our batch-aware contrastive loss during adaptation.

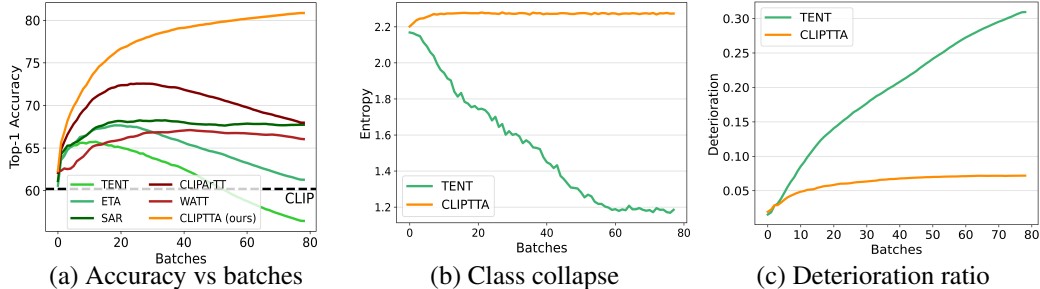

(a) Accuracy vs batches      (b) Class collapse      (c) Deterioration ratio

Figure 4: **CLIPTTA accuracy and robustness on CIFAR-10-C.** (a) In the non-episodic setting, CLIPTTA steadily improves top-1 accuracy across batches while competing methods degrade. (b) CLIPTTA maintains high prediction entropy, preserving diversity in predicted classes, whereas TENT shows marked entropy collapse. (c) The deterioration ratio, defined as the fraction of initially correct predictions that become incorrect, increases significantly for TENT but remains low for CLIPTTA.

## 5   Conclusion

This work introduces CLIPTTA, showing that using a simple soft contrastive loss can be highly beneficial to adapt VLMs in pseudo-label TTA. By a careful analysis of our loss and its gradient, we show that our method brings robustness to the class collapse and pseudo-label drift issues. We also introduce a contrastive outlier exposure loss to tackle the open-set TTA setting. Extensive experiments conducted on a wide range of benchmarks demonstrate that our method significantly outperforms previous baselines on both closed-set and open-set adaptation. Ablation experiments and model analyses strengthen the foundations of our contribution. While CLIPTTA shows strong results, it also presents some limitations. First, it focuses on global image–text interactions, which may limit adaptation to fine-grained or localized shifts. Second, our evaluation is restricted to classification tasks, and extending the approach to broader multimodal settings remains an open direction. Finally, like most test-time adaptation methods, CLIPTTA assumes additional computation and time are available at inference, which may not always be practical in real-world deployments. Future work includes investigating more thoroughly the interplay between visual and textual adaptation during test-time optimization, aiming to better exploit cross-modal consistency. Another promising direction is to design gradient-based adaptation strategies that operate efficiently in real-time environments, for instance within embodied or interactive agents.

## Acknowledgment

This work was supported by the French National Research Agency (Agence Nationale de la Recherche, ANR) under grants from project DIAMELEX (ANR-20-CE45-0026) and project RODEO (ANR-24-CE23-5886). It was granted access to the HPC resources of IDRIS under the allocation AD011013370R2 made by GENCI.

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

# CLIPTTA: Robust Contrastive Vision-Language Test-Time Adaptation - Appendix

The appendix is organized as follows. In Appendix A, we present a detailed theoretical analysis of the gradients of the $\mathcal{L}_{\text{CLIPTTA}}$ loss. In Appendix B, we provide additional insights into how $\mathcal{L}_{\text{CLIPTTA}}$ helps mitigate collapse and pseudo-labeling errors. Finally, in Appendix C, we elaborate on the experimental protocol and include additional experimental results.

## A Theoretical Analysis

### A.1 Gradient Analysis

In this section, we provide a detailed analysis of the gradients of the TENT loss $\mathcal{L}_{\text{TENT}}$, CLIP's contrastive loss $\mathcal{L}_{\text{cont.}}$ (*i.e.* using hard pseudo-captions), our soft contrastive loss $\mathcal{L}_{\text{s-cont.}}$, the regularization loss $\mathcal{L}_{\text{reg}}$ and the CLIPTTA loss $\mathcal{L}_{\text{CLIPTTA}}$. Furthermore, we show how, benefiting from the information of other predictions in the batch, both contrastive losses allow to avoid collapse. Finally, when combined with the regularization loss, CLIPTTA allows mitigating the effect of pseudo label errors.

Let's consider a batch of examples $\boldsymbol{x}_1, ..., \boldsymbol{x}_N$, and let's write $N_k$, the number of predictions assigned to the $k^{th}$ class. To simplify the computations, we place ourselves in the case where only the parameters of the visual encoder are updated.

**Gradient of $\mathcal{L}_{\text{TENT}}$.** We recall that the TENT loss writes as follows:

$$\mathcal{L}_{\text{TENT}} = -\sum_{k=1}^{C} q_{ik} \log q_{ik}$$

with $q_{ik}$ the probability of image $\boldsymbol{x}_i$ being classified as class $k$ (see Eq. (1)). The gradient of $\mathcal{L}_{\text{TENT}}$ w.r.t. $\boldsymbol{z}_i$ is:

$$\nabla_{\boldsymbol{z}_i} \mathcal{L}_{\text{TENT}} = -\sum_{k=1}^{C} \nabla_{\boldsymbol{z}_i} q_{ik} \log q_{ik} = -\sum_{k=1}^{C} (1 + \log q_{ik}) \nabla_{\boldsymbol{z}_i} q_{ik}$$

$$= -\sum_{k=1}^{C} (1 + \log q_{ik}) \, q_{ik} \sum_{c=1}^{C} q_{ic} \, (\boldsymbol{z}_t^k - \boldsymbol{z}_t^c) \qquad (10)$$

$$= -\sum_{k=1}^{C} \Big[ \sum_{c=1}^{C} \log \frac{q_{ik}}{q_{ic}} \, q_{ic} \Big] \, q_{ik} \, \boldsymbol{z}_t^k$$

From Eq. (10) we can see that the gradient will always push $\boldsymbol{z}_i$ in the direction of the predicted class $\hat{k}$ because in that case we have $\log \frac{q_{i\hat{k}}}{q_{ic}} > 0, \forall c \neq \hat{k}$. And there is no mechanism allowing to reduce the magnitude of the gradient towards the predicted class even when we are approaching a situation of collapse.

**Gradient of $\mathcal{L}_{\text{cont.}}$.** Using the notation introduced in the main paper, let $\hat{\boldsymbol{t}}_i$ represent the pseudo-caption associated with the example $\boldsymbol{x}_i$ in the batch, and let $p(\hat{\boldsymbol{t}}_j|\boldsymbol{x}_i)$ denote the probability of $\boldsymbol{x}_i$ matching $\hat{\boldsymbol{t}}_j$ within the batch. Specifically, we have:

$$p(\hat{\boldsymbol{t}}_j|\boldsymbol{x}_i) = \frac{e^{\boldsymbol{z}_i^\top \hat{\boldsymbol{z}}_t^j}}{\sum_{l=1}^{N} e^{\boldsymbol{z}_i^\top \hat{\boldsymbol{z}}_t^l}}$$

The unsymmetrized version of CLIP's contrastive loss writes:

$$\mathcal{L}_{\text{cont.}} = \sum_{i=1}^{N} -\log p(\hat{\boldsymbol{t}}_i|\boldsymbol{x}_i) = \sum_{i=1}^{N} -\boldsymbol{z}_i^\top \hat{\boldsymbol{z}}_t^i + \log \Big( \sum_{j=1}^{N} e^{\boldsymbol{z}_i^\top \hat{\boldsymbol{z}}_t^j} \Big) = \sum_{i=1}^{N} -\boldsymbol{z}_i^\top \hat{\boldsymbol{z}}_t^i + \log \Big( \sum_{k=1}^{C} N_k e^{\boldsymbol{z}_i^\top \boldsymbol{z}_t^k} \Big)$$

where $\widehat{z}_t^{\,j}$ is the embedding of the pseudo caption associated with image $x_j$ and $z_t^k$ is the embedding of class $k$. Let's compute the gradient of $\mathcal{L}_{\text{cont.}}$ w.r.t. $z_i$:

$$
\begin{aligned}
\nabla_{z_i}\mathcal{L}_{\text{cont.}} &= -\widehat{z}_t^{\,i} + \frac{\nabla_{z_i}\sum_{k=1}^{C}N_k e^{z_i^\top z_t^k}}{\sum_{c=1}^{C}N_c e^{z_i^\top z_t^c}} = -\widehat{z}_t^{\,i} + \sum_{k=1}^{C}\frac{N_k e^{z_i^\top z_t^k}}{\sum_{c=1}^{C}N_c e^{z_i^\top z_t^c}}\,z_t^k \\
&= -\widehat{z}_t^{\,i} + \sum_{k=1}^{C}\frac{N_k e^{z_i^\top z_t^k}}{\sum_{c=1}^{C}N_c e^{z_i^\top z_t^c}}\frac{\sum_{c=1}^{C}e^{z_i^\top z_t^c}}{\sum_{c=1}^{C}e^{z_i^\top z_t^c}}\,z_t^k \\
&= -\widehat{z}_t^{\,i} + \sum_{k=1}^{C}\frac{N_k\,q_{ik}}{\sum_{c=1}^{C}N_c\,q_{ic}}\,z_t^k \\
&= -\widehat{z}_t^{\,i} + \sum_{k=1}^{C}w_{k,i}\,z_t^k
\end{aligned}
\tag{11}
$$

with $w_{k,i} = \frac{N_k\,q_{ik}}{\sum_{c=1}^{C}N_c\,q_{ic}}$.

From Eq. (11), we observe that CLIP's contrastive loss consistently drives the visual embedding $z_i$ toward the embedding of its predicted class $\widehat{z}_t^{\,i}$, as $w_{k,i}\leq 1$. However, the gradient's magnitude is influenced by the proportion of predictions assigned to the same class within the batch. Specifically, as the system approaches a collapse scenario (i.e., $w_{k,i}\to 1$), the gradient of $\mathcal{L}_{\text{cont.}}$ diminishes and eventually vanishes:

$$
||\nabla_{z_i}\mathcal{L}_{\text{cont.}}|| \underset{w_{k,i}\to 1}{\longrightarrow} 0
\tag{12}
$$

**Gradient of $\mathcal{L}_{\text{s-cont}}$**   The unsymmetrized version of our $\mathcal{L}_{\text{s-cont}}$ loss writes:

$$
\mathcal{L}_{\text{s-cont}} = \sum_{i=1}^{N} -\sum_{j=1}^{N} p(\hat{t}_j|x_i)\log p(\hat{t}_j|x_i)
$$

Let's compute the gradient of $\mathcal{L}_{\text{s-cont}}$ w.r.t. $z_i$:

$$
\begin{aligned}
\nabla_{z_i}\mathcal{L}_{\text{s-cont}} &= -\sum_{j=1}^{N}\nabla_{z_i}[p(\hat{t}_j|x_i)\log p(\hat{t}_j|x_i)] \\
&= -\sum_{j=1}^{N}p(\hat{t}_j|x_i)\nabla_{z_i}\log p(\hat{t}_j|x_i) + \log p(\hat{t}_j|x_i)\nabla_{z_i}p(\hat{t}_j|x_i).
\end{aligned}
$$

Using the fact that $\nabla p = p\nabla\log p$, we have:

$$
\begin{aligned}
\nabla_{z_i}\mathcal{L}_{\text{s-cont}} &= -\sum_{j=1}^{N}p(\hat{t}_j|x_i)\nabla_{z_i}\log p(\hat{t}_j|x_i) + \log p(\hat{t}_j|x_i)p(\hat{t}_j|x_i)\nabla_{z_i}\log p(\hat{t}_j|x_i) \\
&= -\sum_{j=1}^{N}[1 + \log p(\hat{t}_j|x_i)]p(\hat{t}_j|x_i)\nabla_{z_i}\log p(\hat{t}_j|x_i)
\end{aligned}
$$

Now we can use the fact that $\nabla_{z_i} - \log p(\hat{t}_j|x_i) = -\widehat{z}_t^{\,j} + \sum_{k=1}^{C}w_{k,i}\,z_t^k$ based on the computation $\mathcal{L}_{\text{cont.}}$ in Eq. (11). Therefore, we have:

$$\nabla_{z_i} \mathcal{L}_{\text{s-cont}} = \sum_{j=1}^{N} \beta_{i,j} [-\widehat{z}_t^j + \sum_{k=1}^{C} w_{k,i}\, z_t^k] \tag{13}$$

with $\beta_{i,j} = p(\hat{t}_j|x_i)(1 + \log p(\hat{t}_j|x_i))$.

The gradient of $\mathcal{L}_{\text{s-cont}}$ does not solely push the visual embedding toward the predicted class. Instead, it incorporates other predictions within the batch to guide the gradient direction, thereby mitigating the risk of pseudo-labeling errors. However, similar to CLIP's contrastive loss, the gradient diminishes as we approach a collapse scenario. In the case of collapse, where all examples in the batch are predicted to belong to the same class $c$, the following conditions hold: $w_c(x_i) = 1$ and $w_{k,i} = 0, \forall k \neq c$, and $\widehat{z}_t^j = z_t^c \forall j$. Consequently, the term $[-\widehat{z}_t^j + \sum_{k=1}^{C} w_{k,i}\, z_t^k]$ cancels out, leading to a null gradient.

**Binary classification case.** We derive Eq. (7) in the main paper, starting from Eq. (13), and assuming that the classification task comprises two classes $K = \{a, b\}$, with $N = N_a + N_b$ as the total batch size. To build on the intuition of the working mechanisms of our soft contrastive loss, we adopt the case where class $a$ is dominant in the batch (*i.e.*, $N_a \gg N_b$). First, we expand on the second sum term inside Eq. (13), as follows:

$$\sum_{k=1}^{C} w_{k,i} z_i^k = w_{a,i} z_t^a + w_{b,i} z_t^b = \frac{N_a q_{ia}}{N_a q_{ia} + N b q_{ib}} z_t^a + \frac{N_b q_{ib}}{N_a q_{ia} + N_b q_{ib}} z_t^b = \underbrace{\frac{N_a q_{ia} z_t^a + N_b q_{ib} z_t^b}{N_a q_{ia} + N_b q_{ib}}}_{Q} \tag{14}$$

We notice that we can partition the main sum term in Eq. (13) into two sums that account for the $N_a$ samples predicted as class $a$, and the $N_b$ samples predicted as class $b$:

$$
\begin{aligned}
\nabla_{z_i} \mathcal{L}_{\text{s-cont}} &= \sum_{j=1}^{N} \beta_{i,j}[-\hat{z}_t^j + Q] = N_a \beta_{ia}[-z_t^a + Q] + N_b \beta_{ib}[-z_t^b + Q] \\
&= N_a \beta_{ia}[-z_t^a + \frac{N_a q_{ia}}{N_a q_{ia} + N_b q_{ib}} z_t^a + \frac{N_b q_{ib}}{N_a q_{ia} + N_b q_{ib}} z_t^b] \\
&\quad + N_b \beta_{ib}[-z_t^b + \frac{N_a q_{ia}}{N_a q_{ia} + N_b q_{ib}} z_t^a + \frac{N_b q_{ib}}{N_a q_{ia} + N_b q_{ib}} z_t^b] \\
&= N_a \beta_{ia}[\frac{-N_b q_{ib}}{N_a q_{ia} + N_b q_{ib}} z_t^a + \frac{N_b q_{ib}}{N_a q_{ia} + N_b q_{ib}} z_t^b] \\
&\quad + N_b \beta_{ib}[\frac{-N_a q_{ia}}{N_a q_{ia} + N_b q_{ib}} z_t^b + \frac{N_a q_{ia}}{N_a q_{ia} + N_b q_{ib}} z_t^a] \\
&= \beta_{ia} q_{ib} \frac{N_a N_b}{N_a q_{ia} + N_b q_{ib}} (z_t^b - z_t^a) - \beta_{ib} q_{ia} \frac{N_a N_b}{N_a q_{ia} + N_b q_{ib}} (z_t^b - z_t^a) \\
&= [\beta_{i,a} q_{ib} - \beta_{i,b} q_{ia}] \frac{N_a N_b}{N_a q_{ia} + N_b q_{ib}} (z_t^b - z_t^a)
\end{aligned}
\tag{15}
$$

As pointed out, the increasing dominance of class $a$ ($N_b \rightarrow 0$) reduces the gradient to 0, vanishing the negative effect of class collapse.

**Gradient of $\mathcal{L}_{\text{reg}}$.** The regularization loss $\mathcal{L}_{\text{reg}}$ writes as:

$$\mathcal{L}_{\text{reg}} = -\sum_{c=1}^{C} \bar{q}_c \log \bar{q}_c. \tag{16}$$

where $\bar{q}_c$ correspond to the average predicted probability for class $c$ inside the batch.

Let's compute the gradient of $\mathcal{L}_{reg}$ w.r.t. $z_i$:

$$\nabla_{z_i}\mathcal{L}_{reg} = \sum_{k=1}^{C} \nabla_{z_i}\overline{p}_k \log \overline{p}_k = \sum_{k=1}^{C}(1 + \log \overline{p}_k)\,\nabla_{z_i}\overline{p}_k$$

Therefore, we only need to compute $\nabla_{z_i}\overline{p}_k$:

$$\nabla_{z_i}\overline{p}_k = \nabla_{\mathbf{z}_i}\frac{1}{N}\sum_{i=1}^{N} q_{ik} = \frac{1}{N}\nabla_{\mathbf{z}_i}q_{ik} = \frac{1}{N}\nabla_{\mathbf{z}_i}\frac{e^{z_i^{\top}\mathbf{z}_t^k}}{\sum_{j=1}^{C} e^{z_i^{\top}\mathbf{z}_t^j}} = \frac{1}{N}q_{ik}\sum_{j=1}^{C} q_{ij}\,[\mathbf{z}_t^k - \mathbf{z}_t^j]$$

Then we have:

$$
\begin{aligned}
\nabla_{z_i}\mathcal{L}_{reg} &= \frac{1}{N}\sum_{k=1}^{C}(1+\log \bar{q}_k)q_{ik}\sum_{j=1}^{C} q_{ij}(\mathbf{z}_t^k - \mathbf{z}_t^j)\\
&= \frac{1}{N}\sum_{k=1}^{C}[(1+\log \bar{q}_k)q_{ik}\sum_{j\neq k} q_{ij} - q_{ik}\sum_{j\neq k} q_{ij}(1+\log \bar{q}_j)]\,\mathbf{z}_t^k \qquad (17)\\
&= \frac{1}{N}\sum_{k=1}^{C}[\sum_{j=1}^{C} q_{ij}\log\frac{\bar{q}_k}{\bar{q}_j}]q_{ik}\,\mathbf{z}_t^k
\end{aligned}
$$

From Eq. (17), we observe that the gradient is influenced by the ratios $\log\frac{\bar{q}_k}{\bar{q}_j}$, driving it towards the classes that are underrepresented in the batch predictions. The use of the regularization loss in conjunction with our soft contrastive loss creates a powerful combined effect, enabling the effective relabeling of misclassified examples, as discussed in Appendix B.

**Gradient of $\mathcal{L}_{\text{CLIPTTA}}$.** We recall from the main paper that the final CLIPTTA loss combines both $\mathcal{L}_{\text{s-cont}}$ and $\mathcal{L}_{reg}$, thus benefiting both from an enhanced adaptation loss as well a mechanism to combat pseudo-labeling errors (we omit the effect of the memory for simplicity):

$$\mathcal{L}_{\text{CLIPTTA}} = \mathcal{L}_{\text{s-cont}} + \lambda_{\text{reg}}\mathcal{L}_{\text{reg}}, \qquad (18)$$

Therefore the gradient of $\mathcal{L}_{\text{CLIPTTA}}$ writes:

$$\nabla_{z_i}\mathcal{L}_{\text{CLIPTTA}} = \sum_{j=1}^{N}\beta_{i,j}[-\widehat{\mathbf{z}}_t^j + \sum_{k=1}^{C} w_{k,i}\,\mathbf{z}_t^k] + \lambda_{reg}\frac{1}{N}\sum_{k=1}^{C}[\sum_{j=1}^{C} q_{ij}\log\frac{\bar{q}_k}{\bar{q}_j}]q_{ik}\,\mathbf{z}_t^k. \qquad (19)$$

Depending on the composition of the batch, we can see that $\mathcal{L}_{\text{CLIPTTA}}$ will strongly benefit from the contribution of the soft-contrastive loss to provide accurate adaptation, or will be able to correct misclassified examples due to the positive interaction of the combined corrective terms in $\nabla_{z_i}\mathcal{L}_{\text{s-cont}}$ and $\nabla_{z_i}\mathcal{L}_{reg}$.

## A.2 Analysis of OCE Loss

As discussed in the main paper, our outlier contrastive exposure (OCE) in Eq. (9) of the main paper is a special case of the intra-class variance minimization:

$$\sigma^2 = \quad p_{\text{id}}\frac{\sum_i^{N} w_i(s_i - \mu_{id})^2}{\sum_{i=1}^{N} w_i} + p_{\text{ood}}\frac{\sum_i^{N}(1-w_i)(s_i - \mu_{ood})^2}{\sum_{i=1}^{N}(1-w_i)} \qquad (20)$$

with $p_{\text{id}} = \frac{1}{N}\sum_i w_i$ and $p_{\text{ood}} = \frac{1}{N}\sum_i (1 - w_i)$. This is further condensed into the loss function Eq. (21):

$$\sigma_{\text{intra}}^2 = p_{\text{id}}\mu_{\text{id}}^2 - p_{\text{ood}}\mu_{\text{ood}}^2 \tag{21}$$

Here, samples from the same distribution might tend to collapse into a single point. An alternative formulation is inter-class variance maximization, as shown in Eq. 22:

$$\sigma_{\text{inter}}^2 = p_{\text{id}}p_{\text{ood}}(\mu_{\text{id}} - \mu_{\text{ood}})^2 \tag{22}$$

The impact of the proportions $p_{\text{id}}$ and $p_{\text{ood}}$ is twofold. First, when neither ID nor OOD samples are detected, the respective proportion nullifies, and the inter-class variance reaches its minimum. On the contrary, an equilibrium can be reached with both $p_{\text{id}} = p_{\text{ood}} = 0.5$, which displays the implicit assumption of equally distributed scores between ID and OOD. We argue that this constraint limits the flexibility of the OOD detection at test time; as the nature of incoming samples is unknown, allowing for a non-uniform distribution in the detection can help filter out less useful samples. Secondly, the product of these probabilities would reduce the scale of the loss, especially compared to the other components of our CLIPTTA framework, which limits its impact on the adaptation of the model. Hence, a fully contrastive metric can attain the same detection objective by diminishing the latter negative effects:

$$\sigma_{\text{inter}}^2 = (\mu_{\text{id}} - \mu_{\text{ood}})^2 \tag{23}$$

## B    Discussion on CLIPTTA's robustness

We further study the properties of CLIPTTA, to expand the insights on the working mechanisms that assist in its success. Initially, the accuracy across batches (see Fig. 1 in the main paper) serves as a straightforward depiction of (a) the general preeminence of CLIPTTA over other methods, particularly entropy-based techniques, and (b) the collapse effect in methods such as TENT. To elaborate on the underlying advantages of our method, we examine the adaptation process more closely, first in a controlled toy example, then using CIFAR-10-C across all of its corruptions.

**Mitigating pseudo-label errors.** In Fig. 5, we present a controlled toy example demonstrating how CLIPTTA effectively mitigates misclassifications. This example features a batch of six samples in a three-class classification problem. It focuses on the gradient orientations of the TENT, CLIPTTA, and regularized CLIPTTA losses for a single misclassified and ambiguous sample. The sample in question exhibits high probabilities for both the predicted and correct labels, indicating low confidence. The gradient of the CLIPTTA loss is directed toward the correct label, working to minimize the difference between the top two probabilities—a behavior further amplified by the regularized CLIPTTA loss. In contrast, TENT prioritizes increasing the highest probability, thereby reinforcing the incorrect prediction.

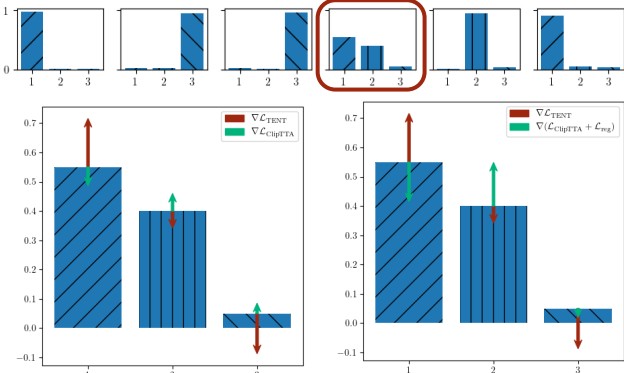

Figure 5: **Gradient Behavior: TENT vs. CLIPTTA** Illustration of gradient directions for TENT, CLIPTTA, and regularized CLIPTTA losses on a misclassified sample (circled in red). While TENT (red arrows) reinforces the incorrect prediction to reduce entropy, CLIPTTA and its regularized version (green arrows) aim to minimize top-2 probability differences, guiding the correction.

We provide quantitative insights on the CIFAR-10-C dataset in Fig. 6. In Fig. 6-a, we observe the collapse of TENT, while CLIPTTA maintains robust performance. To further analyze this, we quantify

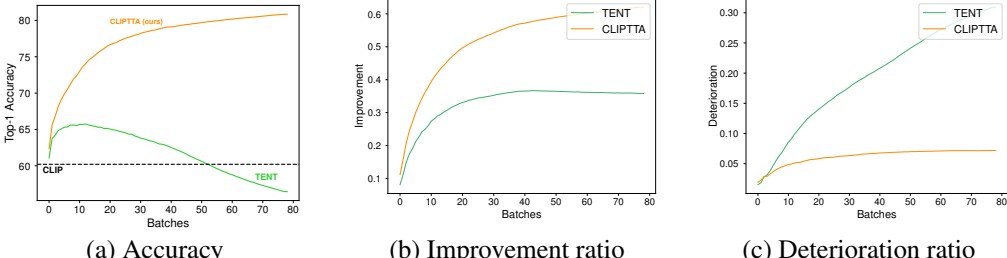

| (a) Accuracy | (b) Improvement ratio | (c) Deterioration ratio |

Figure 6: **Improvement & Deterioration ratios on CIFAR-10-C.** (a) While TENT's accuracy collapses, CLIPTTA shows consistent improvement. (b) The improvement ratio quantifies the proportion of misclassified examples correctly relabeled after adaptation. (c) The deterioration ratio captures the proportion of correctly classified examples that become misclassified post-adaptation.

two key metrics: the "improvement ratio" shown in Fig. 6-b, which represents the proportion of misclassified examples that are correctly classified after adaptation, and the "deterioration ratio" shown in Fig. 6-c, which denotes the proportion of correctly classified examples that become misclassified after adaptation. CLIPTTA outperforms TENT by achieving a higher improvement ratio and a lower deterioration ratio.

## C  Experimental details

We provide further information about the experimental setup that was conducted in the main paper. This includes the specifics of the experimental protocol, the baselines and benchmarks that were considered, as well as an extension of the empirical results.

### C.1  Detailed experimental protocol

In our experiments, we follow the widely explored *non-episodic* TTA setting [11, 17, 28], in which the model is adapted continually to batches of data, without recovering its original weights. This poses a challenge, as adaptation risks of severely degrading the model, which can aggravate as adaptation goes longer. As some of the considered baselines were originally conceived for an *episodic* setting (*e.g.* CLIPArTT [6]), some conditioning was applied in order to amplify their performance in this scenario.

### C.2  Details on baselines and datasets

**Benchmarks.**  We provide more detailed information about the datasets that compose the different benchmarks used through the main paper. For all the experiments, images of different sizes were reshaped for compatibility with CLIP (*i.e.*, to 224×224).

**Natural images.**  We employed CIFAR-10 and CIFAR-100 [31], both containing 10,000 images of size 28×28, and spanning 10 and 100 classes, respectively. We use Imagenet [32] as a larger-scale dataset, with 1000 classes and 50,000 images in total.

**Corruptions.**  Transformed variants of the previous benchmarks are built by applying 15 different corruptions such as *gaussian noise*, *fog*, or *pixelate*. This results in CIFAR-10-C and CIFAR-100-C [33] and Imagenet-C. Each corruption is utilized in its highest severity level (*e.g., level 5*), yielding the most complex version of each dataset. The number of images in each corrupted set and their size correspond to the previous benchmark, which results in 45 different datasets to evaluate in total.

**Fine-grained classification datasets**  are a popular choice in zero-shot classification with CLIP, as they span a wide semantic variety in their classes. We utilize Imagenet as well 10 other datasets covering: Aircraft [34], Caltech101 [35], Cars [36], DTD [37], EuroSat [38], Flowers102 [39], Food101 [40], Pets [41], SUN397 [42], and UCF101 [43]. The specific details of each dataset are condensed in Table 5.

| Dataset | Classes | Size | Category |
|---------|---------|------|----------|
| Aircraft | 100 | 3,333 | Transportation |
| Caltech101 | 100 | 2,465 | Objects |
| Cars | 196 | 8,041 | Transportation |
| DTD | 47 | 1,692 | Textures |
| EuroSat | 10 | 8,100 | Satellite |
| Flowers102 | 102 | 2,463 | Flora |
| Food101 | 101 | 30,300 | Food |
| Pets | 37 | 3,669 | Fauna |
| SUN397 | 397 | 19,850 | Scenes |
| UCF101 | 101 | 3,783 | Actions |

Table 5: **Detailed information of the fine-grained classification benchmark.**

| Dataset | Domain | Size |
|---------|--------|------|
| PACS | Photo | 1,670 |
| | Cartoon | 2,344 |
| | Sketch | 3,929 |
| | Art painting | 2,048 |
| OfficeHome | Art | 965 |
| | Clipart | 2,535 |
| | Product | 2,470 |
| | Real world | 1,495 |

(a) PACS and OfficeHome

| Dataset | Domain | Classes | Size |
|---------|--------|---------|------|
| Imagenet | Natural | 1,000 | 50,000 |
| Imagenet-V2 | Natural | 1,000 | 10,000 |
| Imagenet-S | Sketch | 1,000 | 50,000 |
| Imagenet-R | Art | 200 | 30,000 |
| Imagenet-A | Adversarial | 7,500 | 7,500 |

(b) Imagenet-Domains

Table 6: **Detailed dataset statistics.** (a) PACS and OfficeHome. (b) Imagenet-Domains.

**Domain generalization.** This is a set of datasets popularly use in the context of Domain Adaptation. We use Visda-C [44], which includes 12 common classes and contains two main sets: a set of 152,397 3D renderings and a set of 55,388 of images cropped from MS COCO [45]. We also incorporate PACS [46], with seven classes, and OfficeHome [47] with 65 classes, which include images in four different styles, as summarized in Table 6a). Finally, we include the challenging Imagenet-Domains benchmark, involving four variants of Imagenet: Imagenet-V2 [48], Imagenet-R [49], Imagenet-S [50], Imagenet-A [51], each of which is detailed in Table 6b).

**Out-of-distribution datasets.** In our open-set TTA setup, each ID dataset in the natural and corrupted image benchmarks is paired with a corresponding OOD dataset. The classification task is performed only on ID samples, while OOD samples are solely used for detection (*i.e.*, recognizing and rejecting unknowns). Thus, OOD class labels are not meaningful in this context. Following prior work [28, 27], we use SVHN [52] (26,032 street view digit images) as the OOD set for CIFAR-10 and CIFAR-100, and Places365 [53] (1.8M scene images) for ImageNet. In the corrupted setting (*i.e.*, CIFAR-10/100-C and ImageNet-C), we use SVHN-C and Places365-C as OOD sources, matched by corruption type (*e.g.*, JPEG compression) and set to maximum severity.

**Baselines.** We group baselines into three categories based on their adaptation strategy. The first group includes entropy-based methods for standard classifiers such as TENT [11], ETA [12], SAR [13], RoTTA [14], OSTTA [17], SoTTA [26], STAMP [27], and UniEnt [28]. These methods typically operate by minimizing the conditional entropy of the model's predictions and require adaptations to work with CLIP's vision-language outputs. The second group comprises CLIP-specific methods such as CLIPArTT [6] and WATT [7], which modify the loss or prompt structure to better leverage CLIP's multimodal nature. The third group includes alternative CLIP-based adaptation approaches: TPT [3], which performs prompt tuning via entropy minimization, and TDA [4], which operates without gradients using a memory-based episodic scheme. All baselines are implemented following their respective publications. For CLIP-based methods, minimal changes were needed to integrate into our framework. For non-CLIP methods, we use CLIP's image-to-text similarities (as

defined in Eq.1, Sec.3) as classification logits. Entropy-based baselines directly apply their loss to these logits. Hyperparameter details are provided below when applicable.

- ETA: a similarity threshold of $\epsilon = 1$ and an entropy threshold $\alpha = 0.4$ are used. These are kept for all cases.

- SAR: an entropy threshold $\alpha = 0.4$ and an exponential moving average (EMA) weight $m = 0.2$ are used for all cases. The SAM optimizer is employed.

- RoTTA: we use a timeliness weight $\lambda_t = 1$ and an uncertainty weight $\lambda_u = 1$, a memory capacity equivalent to the batch size. These are kept for all cases.

- TDA: we use the same values used for Imagenet in the original paper. We employ $\alpha_{\mathrm{pos}} = 2.0$, $\beta_{\mathrm{pos}} = 2.0$, $\alpha_{\mathrm{neg}} = 0.117$, $\beta_{\mathrm{neg}} = 1.0$, entropy thresholds $H_o = \{0.2, 0.5\}$, entropy masks $M_o = \{0.03, 1.0\}$, and positive and negative shot capacities of 2 and 3, respectively.

- CLIPArTT: we take $K = 3$ most probable classes in all datasets, except for $K = 5$ in VisDA-C, which uses a learning rate of $1 \times 10^{-5}$.

- WATT: we use two adaptation iterations per text prompt, and two meta-repetitions are used. A learning rate of $1 \times 10^{-5}$ is used for VisDA-C.

- SoTTA: we use the confidence threshold $\tau = 1/|\mathcal{C}|$, with $\mathcal{C}$ the number of classes. The memory capacity is equal to the batch size. The SAM optimizer is employed.

- UniEnt: we use $\lambda_{\mathrm{reg}} = 1$ and $\lambda_{\mathrm{ood}} = 1$.

## C.3 Extended experimental results

**Adapting the text encoder.** CLIPTTA is evaluated across a diverse set of datasets by adapting not only the visual encoder but also the text encoder, as shown in Table 7. Updating the text encoder proves beneficial in many cases, particularly for semantically complex datasets where CLIP's pre-trained embeddings may lack sufficient specialization. This is evident in datasets focused on fine-grained classification, such as SUN397 and OxfordPets, where incorporating text encoder updates yields notable improvements. However, updating the text encoder can sometimes have detrimental effects, especially on datasets containing general or well-represented concepts, such as EuroSat. Despite being visually challenging, the broad and commonly encountered class labels in such datasets may already be adequately represented in CLIP's original text embeddings. In these cases, further adaptation of the text encoder may disrupt

| Dataset | CLIPTTA (Vision only) | CLIPTTA (Vision + Text) |
|---|---|---|
| CIFAR-10 | 95.0 | 93.5 (-1.5) |
| CIFAR-100 | 74.9 | 75.0 (+0.1) |
| ImageNet | 69.1 | 69.6 (+0.5) |
| ImageNet-V2 | 62.7 | 63.1 (+0.4) |
| ImageNet-A | 54.0 | 54.2 (+0.2) |
| ImageNet-R | 80.1 | 79.9 (-0.2) |
| ImageNet-S | 50.8 | 51.2 (+0.4) |
| Aircraft | 26.5 | 26.9 (+0.4) |
| Caltech101 | 94.2 | 94.4 (+0.2) |
| Cars | 66.7 | 67.1 (+0.4) |
| DTD | 46.5 | 48.1 (+1.6) |
| EuroSat | 80.3 | 72.9 (-7.4) |
| Flowers102 | 71.3 | 71.7 (+0.4) |
| Food101 | 86.7. | 86.8 (+0.1) |
| OxfordPets | 91.6 | 92.4 (+0.8) |
| SUN397 | 65.2 | 67.5 (+2.5) |
| UCF101 | 69.3 | 70.3 (+1.0) |
| Median | 69.2 | 70.3 (+1.1) |

Table 7: **Impact of updating the text encoder**.

this alignment, leading to performance degradation. This behavior underscores the importance of selectively adapting the text encoder based on the semantic complexity of the dataset.

Moreover, the results highlight the trade-off between generalization and specialization when jointly adapting both encoders. While semantically complex datasets benefit from increased specialization, datasets with simpler or well-represented class concepts risk losing the robust generalization capabilities inherent to CLIP's pre-trained representations. This suggests that a targeted or dataset-specific strategy for adapting the text encoder may be more effective in leveraging its potential.

**Open-set TTA on corrupted datasets.** Table 9 reports results in the challenging open-set setting under corruption shifts. This scenario is challenging because models must adapt to noisy in-distribution samples while maintaining robustness to unseen OOD classes. As previously observed, TENT is highly unstable in these settings, suffering from severe model collapse that is exacerbated by corrupted inputs. Its accuracy drops to 2.1% on ImageNet-C and 10.6% on CIFAR-100-C, with poor OOD detection (FPR95 above 95%), confirming its sensitivity to pseudo-label noise.

In contrast, CLIPTTA with the OCE loss maintains high performance across all benchmarks, achieving the best overall results on both accuracy and OOD detection. On average, on the corrupted datasets, it improves over UniEnt by +5.8 points in accuracy and reduces FPR95 by nearly 20 points. These gains demonstrate the benefit of aligning the adaptation objective with CLIP's pre-training loss while integrating an explicit OOD detection signal. The results confirm that CLIPTTA is well-suited for open-set test-time adaptation, even under strong distribution shifts such as corruptions.

| | CIFAR-10 | | | CIFAR-100 | | | ImageNet | | | Average | | |
|---|---|---|---|---|---|---|---|---|---|---|---|---|
| | ACC↑ | AUC↑ | FPR95↓ | ACC↑ | AUC↑ | FPR95↓ | ACC↑ | AUC↑ | FPR95↓ | ACC↑ | AUC↑ | FPR95↓ |
| CLIP | 89.3 | 98.5 | 5.2 | 68.1 | 86.8 | 83.5 | 66.7 | 90.1 | 43.8 | 74.7 | 91.8 | 44.2 |
| TENT [11] | 93.0 | 42.3 | 89.3 | 69.1 | 36.2 | 94.8 | 12.4 | 49.9 | 89.4 | 58.2 | 42.8 | 91.2 |
| OSTTA [17] | 90.9 | 60.5 | 72.8 | 70.9 | 43.3 | 93.8 | 66.9 | 84.9 | 59.2 | 76.2 | 62.9 | 75.3 |
| SoTTA [26] | 89.5 | 98.5 | 4.9 | 68.9 | 88.5 | 76.3 | 66.7 | 89.3 | 47.1 | 75.0 | 92.1 | 42.8 |
| STAMP [27] | 89.9 | 98.6 | 5.5 | 67.5 | 87.7 | 80.0 | 29.7 | 63.0 | 80.2 | 62.4 | 83.1 | 55.2 |
| UniEnt [28] | 94.2 | **99.9** | **0.0** | 72.7 | 97.8 | 8.7 | 65.2 | 95.4 | 17.1 | 77.3 | 97.7 | 8.6 |
| CLIPTTA + OCE (ours) | **94.6** | 99.8 | 0.4 | **74.9** | **98.4** | **7.6** | **67.6** | **97.7** | **9.7** | **79.0** | **98.6** | **5.9** |

Table 8: **Open-set TTA results**. Top-1 accuracy with ViT-B/16 backbone on the open-set setting.

| | CIFAR-10-C | | | CIFAR-100-C | | | Imagenet-C | | | Average | | |
|---|---|---|---|---|---|---|---|---|---|---|---|---|
| | ACC↑ | AUC↑ | FPR95↓ | ACC↑ | AUC↑ | FPR95↓ | ACC↑ | AUC↑ | FPR95↓ | ACC↑ | AUC↑ | FPR95↓ |
| CLIP | 60.2 | 88.0 | 58.0 | 35.2 | 67.0 | 93.8 | 24.6 | 68.6 | 89.2 | 40.0 | 74.5 | 80.3 |
| TENT [11] | 26.9 | 50.7 | 91.2 | 10.6 | 43.1 | 95.0 | 2.1 | 48.0 | 95.0 | 13.2 | 47.3 | 93.7 |
| OSTTA [17] | 62.7 | 53.4 | 85.2 | 34.5 | 36.3 | 92.6 | 31.2 | 75.1 | 79.8 | 42.8 | 54.9 | 85.9 |
| STAMP [27] | 60.4 | 88.0 | 57.1 | 34.5 | 66.8 | 93.8 | 9.0 | 53.8 | 92.9 | 34.6 | 69.5 | 81.3 |
| UniEnt [28] | 78.7 | 98.6 | **5.7** | 48.9 | 91.0 | 31.0 | 23.6 | 44.8 | 90.8 | 50.4 | 78.1 | 42.5 |
| CLIPTTA + OCE (ours) | **79.1** | **98.7** | 6.1 | **50.4** | **96.7** | **19.2** | **39.0** | **89.0** | **43.2** | **56.2** | **94.8** | **22.8** |

Table 9: **Open-set TTA results on Corrupted Datasets**. Top-1 accuracy with ViT-B/16 backbone on the open-set setting.

**Overall performance across shift types.** To provide a global overview of our evaluation, we summarize in Tab. 10 the average performance of all methods across the four distribution shift categories considered in this work: (i) synthetic corruptions (CIFAR-10/100-C, ImageNet-C), (ii) domain shifts (ImageNet variants such as Sketch and Paintings), (iii) coarse-grained semantic shifts (CIFAR-10, CIFAR-100), and (iv) fine-grained semantic shifts (11 CLIP zero-shot datasets, including ImageNet). This table offers a consolidated view of the results presented in the following sections. As shown, CLIPTTA consistently achieves the best performance across all shift types, with particularly large improvements in challenging settings where the base CLIP model performs poorly. For example, under synthetic corruptions, CLIP reaches 40.3% accuracy on average, while CLIPTTA attains 58.1%, a gain of +17.8 points and 9.1 points above the second-best method. When the base model already performs well—such as in domain shift benchmarks—the improvements are smaller (83.7% vs. 82.8% for TDA). These results suggest that CLIPTTA 's benefits are more pronounced when the initial model accuracy is low. Detailed results for each dataset are provided in the following sections.

**Domain shifts benchmarks.** We provide the extended results of the different domain shifts (Table 1), including Imagenet-Domains (Table 11), VisDA-C (Table 12), OfficeHome (Table 13), and PACS (Table 14). CLIPTTA achieves the best performance across all of these datasets on average, demonstrating great flexibility across domains. Our method also obtains highly competitive results independently in each sub-dataset.

| Method | Corruptions | Domain Shifts | Coarse-Grained | Fine-Grained |
|---|---|---|---|---|
| CLIP | 40.3 | 81.3 | 78.7 | 64.7 |
| TENT [11] | 35.1 | 82.3 | 83.9 | 65.3 |
| ETA [12] | 42.3 | 82.3 | 84.3 | 65.4 |
| SAR [13] | 48.2 | 82.1 | 82.7 | 65.3 |
| RoTTA [14] | 38.7 | 80.9 | 79.0 | 62.1 |
| CLIPArTT [6] | 49.8 | 80.8 | 80.8 | 64.4 |
| WATT [7] | 43.5 | 82.1 | 81.7 | 65.3 |
| TPT [3] | 38.7 | 80.8 | 78.6 | 65.6 |
| TDA [4] | 42.5 | 82.8 | 80.6 | 67.7 |
| CLIPTTA | **58.1** | **83.7** | **85.2** | **69.8** |

Table 10: **Overall performance across shift types.** Average accuracy (%) across four categories of distribution shifts. Detailed per-dataset results are provided in the following sections.

| | ImageNet | ImageNet-A | ImageNet-V2 | ImageNet-R | ImageNet-S | Average |
|---|---|---|---|---|---|---|
| CLIP | 66.7 | 47.8 | 60.8 | 74.0 | 47.8 | 59.4 |
| TPT (NeurIPS '22) | 69.0 | 54.8 | 63.5 | 77.1 | 47.9 | 62.4 |
| TDA (CVPR '24) | 69.5 | **60.1** | **64.7** | **80.2** | 50.5 | **65.0** |
| TENT (ICLR '21) | 66.5 | 51.3 | 60.2 | 79.4 | 43.7 | 60.2 |
| ETA (ICML '22) | 67.4 | 49.2 | 60.9 | 75.3 | 46.8 | 59.9 |
| SAR (ICLR '23) | 66.7 | 51.5 | 60.5 | 79.6 | 44.6 | 60.6 |
| RoTTA (CVPR '23) | 68.4 | 51.2 | 62.5 | 78.1 | 47.8 | 61.6 |
| CLIPArTT (WACV '25) | 67.6 | 50.7 | 61.2 | 76.2 | 47.9 | 60.7 |
| WATT (NeurIPS '24) | 69.0 | 51.1 | 62.5 | 78.1 | 48.2 | 61.8 |
| CLIPTTA (ours) | **69.6** | 54.0 | 62.7 | **80.2** | 50.8 | 63.4 |

Table 11: **Detailed results on the Imagenet-Domains benchmark.**

| | Synthetic 3D | MS COCO | Average |
|---|---|---|---|
| CLIP | 87.2 | 86.7 | 87.0 |
| TPT [3] | 85.5 | 84.5 | 85.0 |
| TDA [4] | 86.6 | 86.5 | 86.5 |
| TENT [11] | **93.2** | 85.3 | 89.3 |
| ETA [12] | 91.1 | 85.4 | 88.3 |
| SAR [13] | 88.1 | **87.5** | 87.8 |
| RoTTA [14] | 80.6 | 86.7 | 83.7 |
| CLIPArTT [6] | 82.2 | 86.0 | 84.1 |
| WATT [7] | 88.4 | 87.0 | 87.7 |
| CLIPTTA (ours) | 92.2 | 86.9 | **89.6** |

Table 12: **Detailed results on the two domains of the Visda-C dataset.**

| | Art | Clipart | Product | Real | Average |
|---|---|---|---|---|---|
| CLIP | 83.2 | 68.0 | 89.1 | 89.8 | 82.5 |
| TPT [3] | 82.5 | 66.3 | 88.5 | 89.2 | 81.7 |
| TDA [4] | 83.2 | 68.8 | 89.8 | 90.4 | 83.0 |
| TENT [11] | 84.1 | 68.8 | 90.0 | 90.5 | 83.4 |
| ETA [12] | 84.3 | 70.8 | 90.4 | 90.7 | 84.1 |
| SAR [13] | **84.4** | **70.9** | 89.6 | 90.3 | 83.8 |
| RoTTA [14] | 82.9 | 68.0 | 89.1 | 89.8 | 82.5 |
| CLIPArTT [6] | 82.6 | 68.4 | 87.6 | 89.6 | 82.0 |
| WATT [7] | 83.8 | 69.0 | 90.0 | 90.5 | 83.4 |
| CLIPTTA (ours) | 84.2 | 70.7 | **91.0** | **91.0** | **84.2** |

Table 13: **Detailed results on the four domains of the OfficeHome (OH) dataset.**

|  | Photo | Art | Cartoon | Sketch | Average |
|---|---|---|---|---|---|
| CLIP | 99.9 | 97.4 | 99.1 | 88.1 | 96.1 |
| TPT [3] | 99.5 | 95.3 | 93.9 | 87.2 | 94.0 |
| TDA [4] | **99.9** | 97.5 | 98.9 | 88.1 | 96.1 |
| TENT [11] | 99.8 | 98.0 | 99.2 | 89.1 | 96.6 |
| ETA [12] | 99.8 | 97.9 | **99.3** | 89.8 | 96.7 |
| SAR [13] | **99.9** | 97.5 | 99.1 | 88.2 | 96.2 |
| RoTTA [14] | **99.9** | 93.8 | 98.8 | 88.1 | 95.8 |
| CLIPArTT [6] | 99.5 | 96.9 | 98.3 | 90.4 | 96.2 |
| WATT [7] | **99.9** | 97.6 | 99.2 | 88.4 | 96.2 |
| CLIPTTA (ours) | **99.9** | **98.0** | **99.3** | **92.0** | **97.5** |

Table 14: **Detailed results on the four domains of the PACS dataset.**

|  | CIFAR-10 | CIFAR-100 | Average |
|---|---|---|---|
| CLIP | 89.3 | 68.1 | 78.7 |
| TPT [3] | 89.8 | 67.4 | 78.6 |
| TDA [4] | 91.4 | 69.8 | 80.6 |
| TENT [11] | 94.9 | 72.9 | 83.9 |
| ETA [12] | 94.8 | 73.7 | 84.3 |
| SAR [13] | 92.1 | 73.2 | 82.7 |
| RoTTA [14] | 89.4 | 68.5 | 79.0 |
| CLIPArTT [6] | 88.4 | 73.2 | 80.8 |
| WATT [7] | 92.5 | 70.8 | 81.7 |
| CLIPTTA (ours) | **95.1** | **75.3** | **85.2** |

Table 15: **Closed-set TTA on coarse-grained datasets**. Top-1 accuracy with ViT-B/16 backbone on coarse-grained datasets (CIFAR-10 and CIFAR-100).

|  | ImageNet | Aircraft | Caltech101 | Cars | DTD | EuroSAT | Flowers102 | Food101 | OxfordPets | SUN397 | UCF101 | Average |
|---|---|---|---|---|---|---|---|---|---|---|---|---|
| CLIP | 66.7 | 24.8 | 92.2 | 65.5 | 44.1 | 48.3 | 70.7 | 84.8 | 88.4 | 62.3 | 64.7 | 64.7 |
| TPT [3] | 69.0 | 24.8 | **94.2** | 66.9 | **47.8** | 42.4 | 69.0 | 84.7 | 87.8 | 65.5 | 68.0 | 65.6 |
| TDA [4] | 69.5 | 23.9 | **94.2** | **67.3** | 47.4 | 58.0 | 71.4 | 86.1 | 88.6 | **67.6** | **70.7** | 67.7 |
| TENT [11] | 66.5 | 15.5 | 93.8 | 63.0 | 43.1 | 58.4 | 71.3 | 86.5 | 89.5 | 63.1 | 68.0 | 65.3 |
| ETA [12] | 67.4 | 24.8 | 93.0 | 65.2 | 44.4 | 47.5 | **71.4** | 85.9 | 89.2 | 63.6 | 66.6 | 65.4 |
| SAR [13] | 66.7 | 21.9 | 93.9 | 64.0 | 43.9 | 50.2 | 70.9 | 86.5 | 89.6 | 63.3 | 67.7 | 65.3 |
| RoTTA [14] | 68.4 | 22.3 | 94.0 | 58.1 | 45.2 | 24.2 | 70.5 | 81.6 | 87.0 | 64.9 | 66.8 | 62.1 |
| CLIPArTT [6] | 67.5 | 24.0 | 92.7 | 64.0 | 43.4 | 46.7 | 67.0 | 84.2 | 87.1 | 64.2 | 67.0 | 64.4 |
| WATT [7] | 69.0 | 23.6 | 94.1 | 65.8 | 44.7 | 40.0 | **71.4** | 86.2 | 88.7 | 66.3 | 68.2 | 65.3 |
| CLIPTTA (ours) | **69.6** | **26.5** | **94.2** | 66.7 | 46.5 | **80.3** | 71.3 | **86.7** | **91.6** | 65.2 | 69.3 | **69.8** |

Table 16: **Closed-set TTA on fine-grained datasets**. Top-1 accuracy comparison of CLIPTTA against other TTA methods on a suite of 11 fine-grained datasets.

**Semantic datasets.** We report closed-set adaptation results on both coarse- and fine-grained classification tasks in Tables 15 and 16. On coarse-grained benchmarks (CIFAR-10 and CIFAR-100), CLIPTTA achieves the highest accuracy on both datasets, with a strong average of 85.2%, outperforming all TENT-based and CLIP-based methods, including CLIPArTT and WATT. Notably, it improves over TENT by +0.2 points on CIFAR-10 and +2.4 points on CIFAR-100, and remains significantly ahead of zero-shot CLIP (+6.5 points on average). On fine-grained datasets, CLIPTTA consistently ranks among the top methods, achieving the best average accuracy across the 11 datasets (69.8%).

| Backbone | Dataset | Base | TENT | CLIPTTA |
|----------|---------|------|------|---------|
| CLIP-ResNet50 | ImageNet | 58.2 | 58.0 | **59.8** |
| | CIFAR-10 | 68.7 | 72.2 | **84.0** |
| | CIFAR-100 | 40.6 | 44.7 | **55.8** |
| CLIP-ViT-B/32 | ImageNet | 62.0 | 61.4 | **68.2** |
| | CIFAR-10 | 88.7 | 93.2 | **93.4** |
| | CIFAR-100 | 64.0 | 69.6 | **72.5** |
| CLIP-ViT-L/14 | ImageNet | 73.5 | 74.2 | **74.6** |
| | CIFAR-10 | 95.4 | 97.3 | **97.5** |
| | CIFAR-100 | 75.9 | 82.1 | **82.4** |
| SigLIP-ViT-B/16 | ImageNet | 75.7 | 75.5 | **75.9** |
| | CIFAR-10 | 92.5 | **96.0** | **96.0** |
| | CIFAR-100 | 70.9 | 78.2 | **78.7** |

Table 18: **Evaluation across architectures.** CLIPTTA consistently improves over both the base model and TENT across various CLIP and SigLIP backbones. Results are top-1 accuracies (%).

Despite its simplicity, it performs favorably compared to more complex CLIP-specific methods such as TPT, TDA, and CLIPArTT, which rely on prompt tuning or heuristic loss modifications. CLIPTTA performs particularly well on datasets like EuroSAT (+22.3 over CLIPArTT) and OxfordPets (+4.5 over TDA) while maintaining competitive results on the others. These findings highlight the robustness of our adaptation objective across both coarse- and fine-grained tasks without requiring task-specific tuning or architectural modifications.

**Statistical significance.** We conduct additional runs of CLIPTTA to assess its sensitivity to random initialization, reporting the mean accuracy and 95% confidence interval in Tab. 17. The results indicate that CLIPTTA is

| | CIFAR-10 | CIFAR-100 | Imagenet |
|---|----------|-----------|----------|
| CLIPTTA | $94.9 \pm 0.03$ | $75.3 \pm 0.07$ | $69.1 \pm 0.01$ |

Table 17: Accuracy of CLIPTTA averaged over three random initializations (mean $\pm$ 95% CI).

highly stable, with very low variance across independent runs. The tight confidence intervals (e.g., $\pm 0.01$ on ImageNet) confirm the reliability and reproducibility of the observed performance gains, further supporting the robustness of the method across different datasets.

**Evaluation across architectures.** To assess the generality of CLIPTTA beyond the default ViT-B/16 backbone, we extended our experiments to three additional CLIP variants (ResNet50, ViT-B/32, and ViT-L/14) and another contrastive vision-language model, SigLIP-ViT-B/16. As shown in Tab. 18, CLIPTTA consistently improves over both the base model and TENT across all architectures and datasets. For example, on CLIP-ResNet50, it yields gains of +12.0 and +11.1 points on CIFAR-10 and CIFAR-100, respectively, highlighting robustness across training paradigms. Improvements on SigLIP are smaller, with saturation on easier datasets such as CIFAR-10 (96.0%). We expect larger gains on more challenging corrupted benchmarks. These results confirm that our soft contrastive objective generalizes beyond a specific backbone, though all models here were evaluated with shared hyperparameters. Further tuning—especially adapting the loss to SigLIP's pre-training objective—could yield additional improvements.

**Additional baseline comparisons.** We further evaluate CLIPTTA against two additional baselines for test-time adaptation of vision-language models: RPL [54] and BATCLIP [55]. Both methods extend entropy minimization (as used in TENT) with additional mechanisms to improve robustness. RPL leverages pseudo-labels and a generalized cross-entropy loss to mitigate overconfidence, while BATCLIP adapts both visual and textual encoders using two auxiliary objectives—a projection matching loss aligning visual prototypes with text features, and an inter-class separability loss promoting distinctiveness. As shown in Tab. 19, CLIPTTA surpasses both RPL and BATCLIP across CIFAR-10-C, CIFAR-100-C, and ImageNet-C, demonstrating the effectiveness of our soft contrastive objective compared to more complex adaptation strategies. All results are reported using the CLIP

ViT-B/16 backbone under identical evaluation settings.

| Method | CIFAR-10-C | CIFAR-100-C | ImageNet-C | Average |
|---|---|---|---|---|
| CLIP | 60.2 | 35.2 | 25.5 | 40.3 |
| RPL [54] | 61.5 | 38.5 | 25.1 | 41.7 |
| BATCLIP [55] | 73.9 | 42.1 | 30.7 | 48.9 |
| CLIPTTA | **80.7** | **52.6** | **41.1** | **58.1** |

Table 19: **Comparison with RPL and BATCLIP.** Top-1 accuracy (%) on corrupted benchmarks using the CLIP ViT-B/16 backbone.

**Runtime and memory usage.** We compare the runtime and GPU memory usage of our soft contrastive loss with other gradient-based test-time adaptation methods under identical conditions, using a single A6000 GPU and a batch size of 128. As shown in Tab. 20, CLIPTTA introduces negligible memory overhead compared to TENT and achieves competitive runtime, outperforming several recent approaches such as ETA, SAR, CLIPArTT, and WATT. Although TPT [3] is designed for single-image adaptation, its reliance on test-time augmentations leads to similar memory demands and higher latency. Despite its contrastive formulation, CLIPTTA remains efficient even with small batch sizes: for instance, on CIFAR-10 it attains 93.4% accuracy with a batch size of 1, substantially surpassing TENT (40.3%) and TPT (89.8%) under the same conditions. This demonstrates that CLIPTTA is well suited to low-resource or streaming scenarios where batch size is constrained.

| Method | Time (s) | Memory (Gb) |
|---|---|---|
| CLIP | 11 | 1.2 |
| TENT [11] | 312 | 8.1 |
| ETA [12] | 364 | 8.3 |
| SAR [13] | 705 | 8.1 |
| CLIPArTT [6] | 335 | 8.1 |
| WATT [7] | 1260 | 8.3 |
| TPT [3] | 894 | 9.8 |
| $\mathcal{L}_{\text{s-cont}}$ | **315** | **8.1** |

Table 20: **Runtime and memory usage.** Comparison of adaptation time and GPU memory for gradient-based TTA methods. All results are obtained on a single A6000 GPU with batch size 128.

## C.4 Hyperparameter analysis.

In this section, we evaluate the sensitivity of CLIPTTA to its key hyperparameters: the regularization weight $\lambda_{\text{reg}}$, the OOD loss weight $\lambda_{\text{oce}}$, and the adaptation batch size. Results show that CLIPTTA remains robust across a wide range of values, requiring minimal tuning for strong performance.

**Effect of $\lambda_{\text{reg}}$.** Figure 7 shows the impact of the regularization weight $\lambda_{\text{reg}}$ on CIFAR-100 accuracy. We observe that CLIPTTA is remarkably stable for values ranging from 0.5 to 2.0, with accuracy consistently above 75% in this range. Performance peaks around $\lambda_{\text{reg}} = 1$, which we use as the default. Beyond that, accuracy gradually declines, indicating that overly strong regularization may suppress beneficial updates. Overall, this confirms that CLIPTTA does not require precise tuning of $\lambda_{\text{reg}}$ to perform well and that a wide range of values yields near-optimal performance.

**Effect of $\lambda_{\text{oce}}$.** Table 21 reports the impact of $\lambda_{\text{oce}}$ on ImageNet in the open-set setting. While accuracy stays stable for small values, OOD detection improves substantially: AUROC increases from 93.5% (no OCE) to 97.7% at $\lambda_{\text{oce}} = 1$, and FPR95 drops by 16 points. Performance remains robust in the range [0.25–2], confirming the stability of the OCE loss.

| $\lambda_{\text{oce}}$ | 0 | 0.25 | 0.5 | 1 | 2 | 5 | 10 | 20 | 100 |
|---|---|---|---|---|---|---|---|---|---|
| **Acc** | 67.6 | 67.6 | 67.6 | 67.6 | 67.5 | 67.3 | 66.4 | 64.5 | 56.6 |
| **AUC** | 93.5 | 97.5 | 97.6 | 97.7 | 97.8 | 98.0 | 98.4 | 98.8 | 99.2 |
| **FPR** | 25.7 | 10.1 | 9.8 | 9.7 | 8.8 | 7.8 | 6.3 | 4.7 | 2.3 |

Table 21: **Impact of $\lambda_{\text{oce}}$ on Imagenet**. Effect of $\lambda_{\text{oce}}$ on accuracy and open-set metrics AUROC (AUC) and false positive rate (FPR).

**Effect of batch size.** Table 22 compares the performance of TENT, the standalone soft contrastive objective ($\mathcal{L}_{\text{s-cont}}$), and CLIPTTA on CIFAR-10 for batch sizes ranging from 1 to 512. While CLIPTTA benefits slightly from larger batches, its performance remains stable across all sizes. Remarkably, even in the extreme case of batch size 1, CLIPTTA achieves 93.4% accuracy—far above TENT (40.3 %) and $\mathcal{L}_{\text{s-cont}}$ (89.3 %). This robustness is enabled by the Class-wise Confident Memory (CCM), which stores past confident samples and allows contrastive adaptation even when only a single test example is available.

| **Batch size** | 1 | 2 | 8 | 16 | 32 | 64 | 128 | 256 | 512 |
|---|---|---|---|---|---|---|---|---|---|
| TENT [11] | 40.3 | 89.8 | 92.6 | 94.6 | 94.4 | 94.7 | 94.9 | 94.7 | 94.6 |
| $\mathcal{L}_{\text{s-cont}}$ | 89.3 | 90.1 | 93.6 | 94.7 | 94.9 | 94.9 | 95.0 | 94.9 | 94.9 |
| CLIPTTA | **93.4** | **94.7** | **94.7** | **94.8** | **94.8** | **95.0** | **95.1** | **95.1** | **95.2** |

Table 22: **Effect of batch size on CIFAR-10.** Accuracy (%) for different batch sizes. While CLIPTTA benefits from larger batches, it remains competitive even in the case of 1 image adaptation thanks to the Class-wise Confident Memory (CCM).

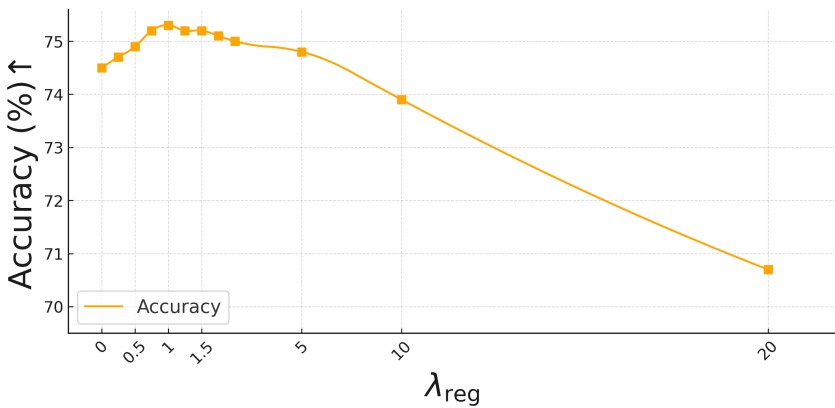

Figure 7: **Impact of $\lambda_{reg}$ on CIFAR-100.** Effect of $\lambda_{reg}$ on the closed-set accuracy of CLIPTTA when evaluated on CIFAR-100.

