# OpenReview forum: "CLIPTTA: Robust Contrastive Vision-Language Test-Time Adaptation"
_NeurIPS.cc/2025/Conference — NeurIPS 2025 poster_

### Official Review · Reviewer_26qe · 2025-06-30

**Clarity:** 3
**Significance:** 3
**Originality:** 3
**Rating:** 4
**Confidence:** 3

**Summary:**

This paper tackles test-time adaptation (TTA) for vision-language models (e.g. CLIP). The authors hypothesize that the classic entropy minimization objective is misaligned with the contrastive training paradigm of CLIP. To address this, they propose a batch-aware contrastive loss for TTA in vision-language settings. Besides, they introduce an outlier contrastive exposure loss to enhance outlier detection in open-set scenarios. Extensive evaluation on 75 datasets verifies robustness and effectiveness of the proposed method for TTA.

**Questions:**

1. How does the number of samples in a batch affect the performance of the proposed contrastive loss for TTA (compare with entropy minimization loss such as TENT)?
2. Though classic models trained with cross-entropy cannot compute the text-to-image entropy (the second term in eq. 3), the image-to-text entropy (the first term in eq. 3) could still be used. What’s the performance when applying the proposed contrastive loss to such classic models? How does it compare to traditional entropy minimization in this context?
3. In Fig. 4 (b), futher ablations such as (1) TENT + $\mathcal{L}\_{\text{reg}}$, and (2) CLIPTTA w/o $\mathcal{L}_{\text{reg}}$ would be helpful to verify the individual contribution of the contrastive loss in preventing class collapse.

**Ethical Concerns:**

["NO or VERY MINOR ethics concerns only"]

**Final Justification:**

My concerns regarding batch size & the analysis of class collapse have been addressed. As for the mismatch between training & adaptation losses, I encourage the authors to refine the motivation to avoid potential misinterpretation. Given most of the issues have been addressed, I will maintain my score, and look forward to improvements in the final version.

**Limitations:**

yes

**Paper Formatting Concerns:**

I do not find major formatting issues in this work.

**Quality:**

3

**Strengths And Weaknesses:**

**Strengths**
- The idea for extending classic entropy minimization to better align with CLIP’s contrastive training is well motivated. The proposed batch-aware contrastive loss is conceptually clear and easy to follow.
- The empirical evaluation is thorough and extensive, spanning a large number of datasets. The improvement shows solid improvements, especially under data corruption, where baseline solutions often fail to adapt.

**Weaknesses**
1. The performance of the proposed batch-aware contrastive loss seems highly depends on batch size (i.e., the number of samples in a batch). However, the effect of batch size on adaptation performance is not studied and discussed.
2. While the authors hypothesize that the misalignment between pre-training and adapting objective is crucial for TTA, this claim remains untested. For example, the performance of applying a similar contrastive loss to classic cross-entropy pre-trained models (e.g., ResNet pre-trained on ImageNet) is unclear. Based on the authors’ hypothesis, will classic entropy minimization techniques outperform the proposed contrastive loss for classic models?
3. Fig. 4b suggests that CLIPTTA avoids class collapse by preserving prediction diversity. However, the regularization term ($\mathcal{L}_{\text{reg}}$), which explicitly maximizes entropy in a batch, likely plays a major role in this effect. It is unclear how much of the improvement in preventing class collapse is due to the proposed contrastive loss itself vs the regularization.

---

> ### Author Rebuttal · Authors · 2025-07-29
>
> We appreciate the reviewer's comments and suggestions, which will help enhance the quality of our work.
>
> 1. __Q1 - Effect of batch size in adaptation:__
> As rightfully noted by the reviewer, the soft contrastive objective $L_{\text{s-cont}}$ requires at least two samples per batch to perform meaningful updates. When the batch size is 1, both the loss and its gradient vanish, resulting in no adaptation. However, as detailed in Section C.4 of the supplementary material, CLIPTTA remains applicable in this setting by leveraging the Class-wise Confident Memory (CCM), which stores past confident samples. This enables the computation of $L\_{\text{s-cont}}$ even when only a single test example is present in the current batch.
>
>     We evaluate the performance of TENT, the standalone soft contrastive loss ($L\_{\text{s-cont}}$), and CLIPTTA across various batch sizes on CIFAR-10. Results are shown in the table below. This comparison highlights that:
>
>     - TENT performs poorly at batch size 1 (40.3%), as the only mechanism it can rely on to prevent reinforcing incorrect predictions in low-batch regimes is through gradient averaging over the batch samples.
>     - The standalone $L_{\text{s-cont}}$ is equivalent to CLIP at batch size 1 (since the loss vanishes), but quickly improves as batch size increases.
>     - CLIPTTA achieves strong performance even with a single test sample (93.4%), thanks to the CCM. At batch size 2, it already surpasses both TENT and $L_{\text{s-cont}}$, and shows only marginal gains beyond batch size 64, indicating robustness to the batch size.
>
>     | Batch size |  1    | 2    | 8    |  16 |  32   | 64   | 128  | 256  | 512  |
>     |:------------:|:-------:|:------:|:------:|:------:|:------:|:------:|:------:|:------:|:------:|
>     | TENT   |  40.3 |  89.8 | 92.6 | 94.6 | 94.4 | 94.7 |94.9| 94.7 | 94.6 |
>     | $L\_{s-cont}$    |  89.3 |  90.1 | 93.6 |  94.7 |  94.9 |  94.9 |  95.0 | 94.9|  94.9|
>     | CLIPTTA   | 93.4 | 94.7 | 94.7  | 94.8 | 94.8 | 95.0 | 95.1 | 95.1 | 95.2 |
>
>     In addition, we compare to TPT and TDA, two recent CLIP-specific TTA methods designed for batch size 1. On CIFAR-10, they achieve 89.8% and 91.4% respectively—substantially below CLIPTTA’s 93.4%, confirming the effectiveness of our memory-based strategy in this regime.
>
>
> 2. __Q2 - Our soft contrastive loss on cross-entropy models:__
> We appreciate the reviewer’s perspective and would like to clarify that our intent was not to suggest that the adaptation loss should necessarily mirror the pre-training objective in all cases. Rather, our motivation for the soft image-text contrastive loss $L_{\text{s-cont}}$ stems from the specific structure of vision-language models (VLMs) like CLIP. This design choice is further supported by insights from the FLYP [1] study, which shows that in few-shot adaptation settings, using a contrastive objective aligned with CLIP’s pre-training yields stronger performance than standard cross-entropy. In our case, the proposed batch-aware adaptation offers two key advantages for CLIP: aligning with the pre-training objective and mitigating pseudo-label errors.
>
>     We agree with the reviewer that, while text-to-image entropy (the second term in Eq. 3) cannot be computed for classic models trained with cross-entropy, the image-to-text entropy (first term in Eq. 3) remains applicable and could be used for adaptation. To fulfill the reviewer’s request, we conducted a preliminary experiment using a ResNet-50 model pre-trained on ImageNet with standard cross-entropy. We compare test-time adaptation using TENT and our proposed $L\_{\text{s-cont}}$, on the ImageNet-C benchmark with Gaussian noise corruption:
>     | Dataset       | ImageNet-C (Gaussian noise) |
>     |:------:|:---------------------------:|
>     | Base    |            20.9             |
>     | TENT    |            21.0             |
>     | $L_{s-cont}$   |            25.0     |
>
>     We observe that $L_{\text{s-cont}}$ yields a +4.1pt gain over the base model, while TENT provides only a marginal +0.1pt improvement. This suggests that the robustness properties of our batch-aware loss—particularly its ability to mitigate pseudo-label drift—may also benefit test-time adaptation of non-VLM models.
>     That said, our main experiments on the same dataset using a CLIP-ViT-B/16 backbone show a much larger improvement: from 13.0% (CLIP base) to 29.0% with CLIPTTA (+16pt). This suggests that part of the gain comes from better robustness, but a substantial component stems from the alignment between the adaptation loss and CLIP’s pre-training objective.
>
> 3. __Q3 - Additional ablations on collapse analysis:__
> To fulfill the reviewer’s request, we conducted an additional ablation study to isolate the contribution of the regularization loss $L_{reg}$ in preventing class collapse during adaptation. The table below reports the evolution of the entropy of the predictions on CIFAR-10-C across batches for: (1) TENT with and without $L_{reg}$, and (2) CLIPTTA with and without $L_{reg}$.
>
>     | Method                                | Batch 0 | Batch 20 | Batch 40 | Batch 60 |
>     |---------------------------------------|:-------:|:--------:|:--------:|:--------:|
>     | TENT                                  | 2.19    | 1.75     | 1.44     | 1.19     |
>     | TENT + $L_{\text{reg}}$               | 2.18    | 2.24     | 2.26     | 2.25     |
>     | CLIPTTA w/o $L_{\text{reg}}$ ($L_{s\text{-}cont}$) | 2.18    | 2.26     | 2.26     | 2.26     |
>     | CLIPTTA ($L_{s\text{-}cont}$ + $L_{\text{reg}}$)   | **2.20**    | **2.27**     | **2.28**     | **2.29**     |
>
>     We agree that the regularization loss $L_{reg}$ plays a beneficial role in preventing class collapse by promoting prediction diversity. However, the results also show that $L_{s\text{-cont}}$ alone (i.e., CLIPTTA w/o $L_{reg}$) achieves nearly identical entropy behavior to TENT + $L_{reg}$, highlighting the inherent robustness of the soft contrastive loss to class collapse. Finally, combining $L_{s\text{-cont}}$ with $L_{reg}$ further improves entropy in predictions, reaching values close to the theoretical maximum (2.3) for CIFAR-10, and confirming their complementarity.
>
> [1] Finetune like you pretrain: Improved finetuning of zero-shot vision models. CVPR 2023

---

> > ### Author Response · Authors · 2025-08-04
> >
> > Dear Reviewer 26qe,
> > Thank you again for your thoughtful review and constructive questions.
> >
> > We hope that our rebuttal has helped clarify the role of batch size in our method, the effect of aligning the adaptation loss with the pre-training objective in the case of CLIP and classical models, and the respective contributions of the contrastive loss and the regularization term to avoiding class collapse.
> >
> > Please do not hesitate to reach out if further clarification would be helpful. We would be glad to discuss any remaining questions you may have.

---

> ### Comment · Reviewer_26qe · 2025-08-05
>
> Thank the author for the detailed experiments. Most of my major concerns have been addressed, including those related to batch size and the analysis of class collapse.
>
> Regarding the mismatch between training & adaptation loss, I would suggest the authors be more careful in presenting this as the primary motivation (e.g., as stated in L28-31). The results clearly show that the proposed contrastive adaptation loss outperforms classic entropy loss, regardless of the underlying models. That is, its effectiveness stems not from mirroring CLIP’s pre-training objective, but simply from being a better alternative to the classic entropy minimization loss.
>
> Given that most of my concerns have been addressed, and with room for improvement in how the motivation is framed, I will maintain my current score. I appreciate the authors’ efforts in answering my concerns, and look forward to seeing improvements in the final version.

---

### Official Review · Reviewer_JXSL · 2025-07-02

**Clarity:** 3
**Significance:** 3
**Originality:** 3
**Rating:** 4
**Confidence:** 4

**Summary:**

In this paper, the authors introduce CLIPTTA, a novel gradient-based test-time adaptation (TTA) method for vision-language models that employs a soft contrastive loss aligned with CLIP’s pre-training objective. They provide a theoretical analysis of CLIPTTA’s gradients, demonstrating how its batch-aware design enhances robustness against pseudo-label drift and class collapse. Extensive experiments validate the effectiveness of the proposed approach.

**Questions:**

1. The authors only consider CLIP in this paper, can the proposed method be applied to other similar vision-language models?

2. How does the proposed method compare with RPL [1] and BATCLIP [2], which are both specifically designed for the same task?

3. While the proposed method may help prevent convergence to dominant classes, it remains unclear how it facilitates recovery from poor pseudo-labeling. Further explanation is needed.


[1] If your data distribution shifts, use self-learning

[2] BATCLIP: Bimodal Online Test-Time Adaptation for CLIP

**Ethical Concerns:**

["NO or VERY MINOR ethics concerns only"]

**Final Justification:**

My concerns are addressed and I keep my current rating.

**Limitations:**

Yes

**Quality:**

3

**Strengths And Weaknesses:**

Strengths:

1. The idea to leverage soft contrastive image-text loss for adapting CLIP is well-motivated.

2. The method is clearly explained and the authors include sufficient analysis to show the benefits of the proposed method.


Weaknesses:

1. Several concepts are introduced without sufficient explanation—for example, gradient dynamics, pseudo-label drift, and natural continuity.

2. It is unclear how the proposed loss function helps minimize the domain gap during test-time adaptation.

3. The proposed method relies on a batch of test samples and cannot be applied to a single test sample.

---

> ### Author Rebuttal · Authors · 2025-07-29
>
> We thank the reviewer for their valuable feedback and insightful questions. Below, we provide detailed responses to each concern raised.
>
> 1) **Q1 - Extension to other similar vision-language models**:
> The reviewer rightly asks whether our method can generalize to other VLMs beyond CLIP.  We extended our experiments during the rebuttal period to assess the generality of CLIPTTA beyond the default ViT-B/16 backbone. Specifically, we evaluated the method on three additional CLIP architectures (ResNet50, ViT-B/32, and ViT-L/14), as well as another contrastive vision-language backbone (SigLIP-ViT-B/16).
>
>     | Backbone         | Dataset    | Base  | TENT  | CLIPTTA (ours) |
>     |------------------|------------|:-----:|:-----:|:-------------------:|
>     | CLIP-ResNet50     | ImageNet   | 58.2  | 58.0  | **59.8**            |
>     |                  | CIFAR-10   | 68.7  | 72.2  | **84.0**            |
>     |                  | CIFAR-100  | 40.6  | 44.7  | **55.8**            |
>     | CLIP-ViT-B/32     | ImageNet   | 62.0  | 61.4  | **68.2**            |
>     |                  | CIFAR-10   | 88.7  | 93.2  | **93.4**            |
>     |                  | CIFAR-100  | 64.0  | 69.6  | **72.5**            |
>     | CLIP-ViT-L/14     | ImageNet   | 73.5  | 74.2  | **74.6**            |
>     |                  | CIFAR-10   | 95.4  | 97.3  | **97.5**            |
>     |                  | CIFAR-100  | 75.9  | 82.1  | **82.4**            |
>     | SigLIP-ViT-B/16   | ImageNet   | 75.7  | 75.5  | **75.9**            |
>     |                  | CIFAR-10   | 92.5  | **96.0**   | **96.0**            |
>     |                  | CIFAR-100  | 70.9  | 78.2  | **78.7**            |
>
>     Overall, CLIPTTA consistently outperforms both the base model and TENT across all architectures and datasets. For instance, on CLIP-ResNet50, CLIPTTA improves performance over TENT by +12 points on CIFAR-10 and +11.1 points on CIFAR-100, demonstrating its effectiveness across diverse architectures and training paradigms. On the other hand, performance gains appear more modest on SigLIP, with saturation observed on some datasets such as CIFAR-10, where both TENT and CLIPTTA reach 96.0%. We anticipate, however, that improvements could be more pronounced on more challenging benchmarks, such as corrupted datasets with lower zero-shot accuracy. This remains an avenue we were unable to explore within the rebuttal timeline.
>
>     These results confirm that our soft contrastive objective’s effectiveness is not tied to a specific backbone or contrastive formulation. We emphasize, however, that these are preliminary results obtained using a shared hyperparameter setting across all models. We anticipate that further tuning—such as adjusting the learning rate for each backbone—could lead to improved performance. Moreover, adapting the adaptation loss to better align with SigLIP’s pre-training objective could further improve performance and is a promising direction for future work.
>
>
> 2) **Q2 - Comparison with RPL[1] and BATCLIP[2]**:
> We thank the reviewer for pointing us to the related works RPL [1] and BATCLIP [2]. Both address the same task—test-time adaptation of VLMs—and extend entropy minimization (as used in TENT) with mechanisms to mitigate its limitations. RPL leverages pseudo-labels and uses a generalized cross-entropy loss to enhance robustness, while BATCLIP adapts both visual and textual encoders using two auxiliary objectives: a projection matching loss to align visual prototypes with text features, and an inter-class separability loss to enforce distinctiveness. In this sense, both remain conceptually close to TENT but aim to address its failure modes.
>
>     In terms of performance, CLIPTTA significantly outperforms both RPL and BATCLIP on CIFAR-10-C, CIFAR-100-C, and ImageNet-C (see table below). These results (from the BATCLIP paper, using ViT-B/16) follow the same setting as ours and enable direct comparison. We will include these comparisons in the final version to highlight the advantages of our simpler soft contrastive loss.
>
>     | Dataset      | CLIP    | RPL   | BATCLIP | CLIPTTA (ours) |
>     |---------------|:-----:|:-----:|:-------:|:----------------:|
>     | CIFAR-10-C    | 60.2| 61.5 | 73.9   | **80.7**         |
>     | CIFAR-100-C   | 35.2| 38.5 | 42.1   | **52.6**         |
>     | ImageNet-C   | 25.5 | 25.1 | 30.7   | **41.1**         |
>
> 3) **Q3 & W2 - On CLIPTTA recovery from poor pseudo-labelling**:
> The ability to recover from poor pseudo-labeling is a subtle yet critical aspect of CLIPTTA's robustness. Unlike methods with explicit correction mechanisms, CLIPTTA relies on the structure of its soft contrastive loss—specifically the $\beta_{i,j}$ coefficients in Eq. (5)—to guide updates.
>
>     This can be understood by contrasting it with TENT: In TENT, the gradient is computed independently for each sample and always reinforces its own current prediction, whether correct or not. Once a sample is misclassified, updates tend to push it further in the wrong direction.
>
>     In contrast, CLIPTTA uses a batch-aware loss where each sample’s gradient is influenced by its similarity to others in the batch via the $\beta_{i,j}$ terms:
>
>     $\beta_{i,j} = p(\hat{t}_j \mid \mathbf{x}_i) \left(1 + \log p(\hat{t}_j \mid \mathbf{x}_i)\right)$,
>
>     where $\hat{t}\_{j}$ is the pseudo-caption of sample $x_j$, and $p(\hat{t}\_{j} | x_{i})$ is the probability that $x_i$ is assigned to $\hat{t}_{j}$.  These coefficients are large when the probability $p(\hat{t}\_j \mid \mathbf{x}\_i) $ that $x_i$ and $x_j$ share the same pseudo-label is high, which typically happens when the two samples are semantically similar. Consider three test samples:
>     - $x_1$ is misclassified as class $c’$.
>     - $x_2$ is confidently and correctly classified as class $c$ (the true class of $x_1$).
>     - $x_3$ is unrelated.
>
>     In TENT, $x_1$ is pulled toward $c’$, reinforcing the mistake. In CLIPTTA, the loss considers the pair ($x_1$, $x_2$). Since $x_1$ assigns a high score to the pseudo-caption of $x_2$, the coefficient $\beta_{1,2}$ is large. This pulls the gradient update for $x_1$ toward the correct class $c$, enabling implicit correction. The unrelated $x_3$ contributes little due to a small $\beta_{1,3}$.
>
>     This behavior is discussed in section 3.2 of the main paper and Appendix A and B. The key idea is that CLIPTTA allows each sample to benefit from other confident and semantically aligned predictions in the batch. This reduces the risk of reinforcing early pseudo-labeling errors.
>     We hope this additional explanation and example help clarify the role of the $\beta_{i,j}$ coefficients in allowing the model to recover from wrong pseudo-labels.
>
> 4) **W1 - Clarification of key concepts**:
> The notions of gradient dynamics (L.31), pseudo-label drift (L.37), and natural continuity (L.48) are meant to provide intuition about the behavior of TTA in the context of CLIP and motivate our loss design. To clarify:
>     - **Gradient dynamics** refer to the direction and magnitude of updates during optimization. When the adaptation loss deviates significantly from the pre-training objective, it may lead to misaligned gradients (in direction and/or magnitude) and suboptimal updates.
>     - **Pseudo-label drift** describes the phenomenon where early adaptation errors (i.e., incorrect pseudo-labels) reinforce themselves over time, gradually pushing the model toward biased or incorrect predictions.
>     - **Natural continuity in adaptation** refers to designing the adaptation process to behave as similarly as possible to the pre-training phase. By aligning the loss structure, we promote stable updates.
>
> 5) **W3 - CLIPTTA on a single test sample**:
> As rightfully noted by the reviewer, the soft contrastive objective $L_{\text{s-cont}}$ requires at least two samples per batch to perform meaningful updates. When the batch size is 1, both the loss and its gradient vanish, resulting in no adaptation. However, as detailed in Section C.4 of the supplementary material, CLIPTTA remains applicable in this setting by leveraging the Class-wise Confident Memory (CCM), which stores past confident samples. This enables the computation of $L\_{\text{s-cont}}$ even when only a single test example is present in the current batch.
>
>     We evaluate the performance of TENT, the standalone soft contrastive loss ($L\_{\text{s-cont}}$), and CLIPTTA across various batch sizes on CIFAR-10. Results are shown in the table below. This comparison highlights that:
>
>     - TENT performs poorly at batch size 1 (40.3%), as the only mechanism it can rely on to prevent reinforcing incorrect predictions in low-batch regimes is through gradient averaging over the batch samples.
>     - The standalone $L_{\text{s-cont}}$ is equivalent to CLIP at batch size 1 (since the loss vanishes), but quickly improves as batch size increases.
>     - CLIPTTA achieves strong performance even with a single test sample (93.4%), thanks to the CCM. At batch size 2, it already surpasses both TENT and $L_{\text{s-cont}}$, and shows only marginal gains beyond batch size 64, indicating robustness to the batch size.
>
>     | Batch size |  1    | 2    | 8    |  16 |  32   | 64   | 128  | 256  | 512  |
>     |:------------:|:-------:|:------:|:------:|:------:|:------:|:------:|:------:|:------:|:------:|
>     | TENT   |  40.3 |  89.8 | 92.6 | 94.6 | 94.4 | 94.7 |94.9| 94.7 | 94.6 |
>     | $L_{s-cont}$    |  89.3 |  90.1 | 93.6 |  94.7 |  94.9 |  94.9 |  95.0 | 94.9|  94.9|
>     | CLIPTTA   | 93.4 | 94.7 | 94.7  | 94.8 | 94.8 | 95.0 | 95.1 | 95.1 | 95.2 |
>
>     In addition, we compare to TPT and TDA, two recent CLIP-specific TTA methods designed for batch size 1. On CIFAR-10, they achieve 89.8% and 91.4% respectively—substantially below CLIPTTA’s 93.4%, confirming the effectiveness of our memory-based strategy in this regime.
>
> [1] If your data distribution shifts, use self-learning. TMLR 2022
>
> [2] BATCLIP: Bimodal Online Test-Time Adaptation for CLIP, arxiv 2024

---

> > ### Author Response · Authors · 2025-08-04
> >
> > Dear Reviewer JXSL,
> > Thank you again for your thoughtful and constructive review.
> >
> > We hope that our rebuttal provided helpful clarifications regarding the conceptual motivations, the correction mechanisms of our contrastive loss, the performance in single-sample settings, and the comparisons with related works such as RPL and BATCLIP.
> >
> > We would gladly provide further clarification should any aspect of our response require more detail.

---

> > > ### Comment · Reviewer_JXSL · 2025-08-08
> > > **Response**
> > >
> > > Thanks for the rebuttal. My concerns are addressed and I keep my current rating.

---

### Official Review · Reviewer_4cbr · 2025-07-03

**Clarity:** 3
**Significance:** 2
**Originality:** 3
**Rating:** 4
**Confidence:** 5

**Summary:**

This paper introduces CLIPTTA, a test-time adaptation method for vision-language models like CLIP. It uses CLIP-style contrastive loss and a soft OOD filtering mechanism based on maximum softmax scores. A learnable threshold and optional memory bank further improve adaptation. Experiments show considerable performance in both closed- and open-set settings.

**Questions:**

Please refer to weaknesses

**Ethical Concerns:**

["NO or VERY MINOR ethics concerns only"]

**Final Justification:**

Thank you for your response and the additional experiments. I have raised my score accordingly. However, since your work focuses on CLIP adaptation, restricting the evaluation to the standard TTA setting is somewhat limited. I strongly recommend including the new experiments in W3 that address cross-dataset adaptation rather than solely corruption-based shifts.

Regarding your response to W2, there are indeed other works proposing training-free adaptation, such as TDA and TCA, which perform adaptation using augmented or single-batch data. This makes your claim of uniqueness less convincing. Please include a discussion of these related works in the final version.

Overall, the paper presents strong experimental results. I recommend some minor refinement in your final version.

**Limitations:**

yes

**Quality:**

3

**Strengths And Weaknesses:**

__Strengths:__

This paper presents a well-structured study on test-time adaptation for CLIP, evaluated across a wide range of datasets. The method demonstrates consistently strong performance, highlighting its robustness and general applicability.

__Weaknesses:__

1. The definition of ID/OOD is somewhat unclear for vision-language models like CLIP, given its undisclosed training data. A sample flagged as OOD by confidence-based filtering may simply reflect underrepresentation, not a true open-set case. A more formal definition would strengthen the framing.

2. The adaptation is performed over 10 steps per sample, which raises questions about test-time efficiency in real-time settings.

3. The method is evaluated solely on CLIP, without examining other vision-language backbones (e.g., SigLIP) or standard CLIP backbones (e.g., ResNet, ViT-L), which limits the generality.

4. The ablation is restricted to typical TTA benchmarks and does not include cross-dataset evaluations commonly used in CLIP adaptation work. Comparing with non-CLIP-specific methods like Tent may also be suboptimal.

5. The paper allows the CLIP logit scale parameter to be updated during test-time adaptation but does not analyse or report its behaviour. Given that CLIP’s pre-trained logit scale is calibrated for contrastive learning, adapting it at test time could introduce instability or overfitting.

---

> ### Author Rebuttal · Authors · 2025-07-29
>
> We thank the reviewer for the thoughtful and constructive feedback, which helped us identify areas for clarification and improvement. Below, we respond to each point in detail, including additional experiments and clarifications where appropriate.
>
> 1) **W1 - Definition of ID/OOD in the CLIP setting**: While the notion of in-distribution (ID) versus out-of-distribution (OOD) may appear ambiguous for vision-language models like CLIP, we follow the task-specific definition of ID/OOD separation as introduced in the MCM paper [1], as mentioned in Section 3.3 of the submission. To clarify, in this context, ID refers to the downstream classification task (e.g., ImageNet-1k or CIFAR-100), which defines the relevant label space, while OOD refers to semantically unrelated samples outside that space. Following [1], this setup has been widely adopted in the CLIP-based OOD detection community [2–4], and we will be glad to clarify and give more details in the related work section.
>
>     Since our objective is to adapt CLIP to a specific task at test time, it is essential to avoid updating the model based on inputs that lie outside the downstream label space. For example, when adapting to ImageNet-1k, including samples from unrelated datasets such as Places or SUN would deteriorate the adaptation process. Our confidence-based filtering addresses this by discarding low-confidence predictions unlikely to belong to the ID distribution.
>     We agree that low-confidence inputs may include underrepresented or ambiguous ID examples. However, such samples also carry a higher risk of adaptation errors. Excluding them is often beneficial in practice, especially in the absence of labels. To improve filtering beyond static thresholds, we introduce the OCE loss, which learns to better distinguish ID from OOD samples during test-time, enhancing adaptation in open-set scenarios.
>
> 2) **W2 - Test-time efficiency**:
> We follow the standard practice in gradient-based TTA, where adaptation is performed using 10 gradient steps [5-7]. This setup provides a good balance between adaptation effectiveness and computation cost, and allows for a controlled and fair comparison with existing baselines. Importantly, our method remains efficient in practice. As detailed in our response to Reviewer 7qJh (Q1), our soft contrastive loss performs adaptation in less time and with minimal memory overhead compared to other competitive TTA methods.
>
> 3) **W3 - Limited backbone evaluation**:
> As requested by the reviewer, we extended our experiments during the rebuttal period to assess the generality of CLIPTTA beyond the default ViT-B/16 backbone. Specifically, we evaluated the method on three additional CLIP architectures (ResNet50, ViT-B/32, and ViT-L/14), as well as another contrastive vision-language backbone (SigLIP-ViT-B/16).
>
>     | Backbone         | Dataset    | Base  | TENT  | CLIPTTA (ours) |
>     |------------------|------------|:-----:|:-----:|:-------------------:|
>     | CLIP-ResNet50     | ImageNet   | 58.2  | 58.0  | **59.8**            |
>     |                  | CIFAR-10   | 68.7  | 72.2  | **84.0**            |
>     |                  | CIFAR-100  | 40.6  | 44.7  | **55.8**            |
>     | CLIP-ViT-B/32     | ImageNet   | 62.0  | 61.4  | **68.2**            |
>     |                  | CIFAR-10   | 88.7  | 93.2  | **93.4**            |
>     |                  | CIFAR-100  | 64.0  | 69.6  | **72.5**            |
>     | CLIP-ViT-L/14     | ImageNet   | 73.5  | 74.2  | **74.6**            |
>     |                  | CIFAR-10   | 95.4  | 97.3  | **97.5**            |
>     |                  | CIFAR-100  | 75.9  | 82.1  | **82.4**            |
>     | SigLIP-ViT-B/16   | ImageNet   | 75.7  | 75.5  | **75.9**            |
>     |                  | CIFAR-10   | 92.5  | **96.0**   | **96.0**            |
>     |                  | CIFAR-100  | 70.9  | 78.2  | **78.7**            |
>
>     Overall, CLIPTTA consistently outperforms both the base model and TENT across all architectures and datasets. For instance, on CLIP-ResNet50, CLIPTTA improves performance over TENT by +12 points on CIFAR-10 and +11.1 points on CIFAR-100, demonstrating its effectiveness across diverse architectures and training paradigms. On the other hand, performance gains appear more modest on SigLIP, with saturation observed on some datasets such as CIFAR-10, where both TENT and CLIPTTA reach 96.0%. We anticipate, however, that improvements could be more pronounced on more challenging benchmarks, such as corrupted datasets with lower zero-shot accuracy. This remains an avenue we were unable to explore within the rebuttal timeline.
>
>     These results confirm that our soft contrastive objective’s effectiveness is not tied to a specific backbone or contrastive formulation. We emphasize, however, that these are preliminary results obtained using a shared hyperparameter setting across all models. We anticipate that further tuning—such as adjusting the learning rate for each backbone—could lead to improved performance. Moreover, adapting the adaptation loss to better align with SigLIP’s pre-training objective could further improve performance and is a promising direction for future work.
>
> 4) **W4 - Limited ablation and baselines**:
> We have already evaluated our method on the 11-dataset cross-dataset benchmark commonly used in CLIP adaptation works in Figure 3(b) of the main paper, with detailed results provided in supplementary material C.3 Table 15. As requested, we provide additional ablations on this benchmark using the soft contrastive loss Lscont, its regularized variant, and memory-enhanced adaptation. As shown in the table below, $L_{s-cont}$ accounts for the vast majority of the overall performance
> gains over entropy minimization (TENT), and incorporating the regularization loss and the CCM memory further boosts performance.
>
>     | Method | Average Accuracy |
>     | :---- | :---: |
>     | CLIP | 64.8 |
>     | TENT | 65.1 |
>     |$L_{s-cont}$| 68.3 |
>     | $L_{s-cont}$ \+ $L_{reg}$| 69.8 |
>     |$L_{s-cont}$ \+  $L_{reg}$ \+ $M$ | 69.9 |
>
>     Regarding the relevance of baselines such as TENT, we emphasize that the primary goal of this paper is to address the question: **What is the best loss function for gradient-based test-time adaptation of CLIP?** This motivates our evaluation protocol, which focuses on comparisons with both generic gradient-based methods like TENT and CLIP-specific gradient-based methods, including CLIPArTT [6] and WATT [7].
>
>     In addition, we include comparisons with other state-of-the-art CLIP-specific methods such as TPT [8] and TDA [9], as reported in Table 2 of the main paper. These results show that CLIPTTA outperforms all prior methods on most datasets. Notably, while TDA excels on the cross-dataset benchmark, it performs worse than the frozen CLIP model on others, such as VisDA-C, highlighting its lack of robustness across shifts. In contrast, CLIPTTA maintains strong performance across all evaluated datasets. Please see our response to Reviewer 7qJh (Q2 & W3) for a detailed breakdown of performance across different types of distribution shifts, and see our response to Reviewer JXSL (Q2) for additional comparisons with two other baselines (RPL[10] and BATCLIP [11]).
>
> 5) **W5 - Logit scale parameter adaptation**:
> We clarify that CLIP’s softmax temperature (logit scale) is not updated during test-time adaptation, in our method or any of the baselines evaluated in this paper. We will make this explicit in the final version.
>
> [1] Delving into Out-of-Distribution Detection with Vision-Language Representations. NeurIPS 2022
>
> [2] LoCoOp: Few-Shot Out-of-Distribution Detection via Prompt Learning. NeurIPS 2023
>
> [3] GL-MCM: Global and Local Maximum Concept Matching for Zero-Shot Out-of-Distribution Detection. IJCV 2025
>
> [4] Negative Label Guided OOD Detection with Pretrained Vision-Language Models. ICLR 2024
>
> [5] Tent: Fully Test-Time Adaptation by Entropy Minimization. ICLR 2021
>
> [6] CLIPArTT: Adaptation of CLIP to New Domains at Test Time. WACV 2025
>
> [7] WATT: Weight Average Test Time Adaptation of CLIP. NeurIPS 2024
>
> [8] Test-Time Prompt Tuning for Zero-Shot Generalization in Vision-Language Models. NeurIPS 2022
>
> [9] Efficient Test-Time Adaptation of Vision-Language Models. CVPR 2024
>
> [10] If your data distribution shifts, use self-learning. TMLR 2022
>
> [11] BATCLIP: Bimodal Online Test-Time Adaptation for CLIP, arxiv 2024

---

> > ### Author Response · Authors · 2025-08-04
> >
> > Dear Reviewer 4cbr,
> >
> > Thank you again for your detailed and constructive review.
> >
> > We hope that our rebuttal clarified the main concerns you raised, including the definition of ID/OOD in the CLIP setting, the efficiency of the method at test-time, the generalization across backbones and datasets, as well as the evaluation protocol and logit scale handling.
> >
> > We would gladly provide further clarification and are happy to continue the conversation if any aspect of our response requires more detail.

---

> > > ### Comment · Reviewer_4cbr · 2025-08-06
> > >
> > > Thank you for your response and the additional experiments. I have raised my score accordingly. However, since your work focuses on CLIP adaptation, restricting the evaluation to the standard TTA setting is somewhat limited. I strongly recommend including the new experiments in W3 that address cross-dataset adaptation rather than solely corruption-based shifts.
> > >
> > > Regarding your response to W2, there are indeed other works proposing training-free adaptation, such as TDA and TCA, which perform adaptation using augmented or single-batch data. This makes your claim of uniqueness less convincing. Please include a discussion of these related works in the final version.
> > >
> > > Overall, the paper presents strong experimental results. I recommend some minor refinement in your final version.

---

### Official Review · Reviewer_7qJh · 2025-07-03

**Clarity:** 3
**Significance:** 2
**Originality:** 3
**Rating:** 5
**Confidence:** 4

**Summary:**

This paper presents CLIPTTA, a novel test-time adaptation (TTA) framework tailored for vision-language models such as CLIP. CLIPTTA addresses the limitations of entropy-based adaptation objectives by introducing a soft contrastive image-text loss, which better aligns with CLIP’s original contrastive pretraining. This loss helps reduce pseudo-label drift and mitigates class collapse during adaptation. Additionally, the paper proposes an Outlier Contrastive Exposure (OCE) loss, aimed at improving in-distribution (ID) and out-of-distribution (OOD) separation for more robust adaptation under distribution shifts. CLIPTTA is evaluated across an extensive benchmark of 75 datasets and shows consistent improvements over standard entropy-based approaches (e.g., TENT) and is competitive with recent state-of-the-art methods such as CLIPArTT.

**Questions:**

1) Could the authors provide a runtime and memory usage comparison with other TTA methods? Given the reliance on large batch sizes for contrastive loss, how does CLIPTTA scale to lower-resource settings?

2) Can the authors provide more detailed per-dataset or per-shift-type analysis? Are there specific types of distribution shifts (e.g., adversarial, synthetic corruptions) where CLIPTTA struggles or particularly excels?

3) In Figure 4, how does CLIPTTA compare to baselines in terms of class collapse and deterioration ratio? Could this be discussed more explicitly?

4) How does the Class-wise Confident Memory (CCM) handle OOD samples during updates? Are there safeguards to prevent incorporating noisy or low-confidence predictions?

5) Does optimizing the OCE loss introduce a trade-off in classification performance on in-distribution samples? If so, how significant is this trade-off?

**Ethical Concerns:**

["NO or VERY MINOR ethics concerns only"]

**Final Justification:**

The authors have addressed most of my concerns during this rebuttal period. While there are some clarity issues and missing discussion regarding the limitation of the work, I think that these issues can be fixed in the camera ready.

**Limitations:**

Missing limitations and not properly discussed in the conclusion.

**Quality:**

3

**Strengths And Weaknesses:**

Strengths:

1) The introduction of the soft contrastive loss is a principled design choice, closely aligned with CLIP’s pretraining, and serves as a more appropriate alternative to entropy minimization for VLMs.

2) The proposed OCE loss improves robustness by explicitly encouraging separation between ID and OOD samples, which is critical for real-world generalization.

3) The paper is clearly written and well-structured, with strong empirical results across a diverse and challenging set of benchmarks.

Weaknesses:

1) Computational Efficiency and Scalability:
The method’s reliance on large batch sizes for effective contrastive loss raises concerns about efficiency and scalability. Although the paper mentions using two NVIDIA V100 GPUs, it does not report runtime, memory usage, or how performance scales with smaller hardware setups. A quantitative analysis of resource requirements would strengthen the work.

2) Unexplored Failure Cases of Contrastive Loss:
The paper does not examine scenarios where soft contrastive loss might break down—e.g., when pseudo-labels are highly correlated within a batch, or when the batch contains a high proportion of OOD samples. Such conditions could undermine the contrastive objective, and identifying these limits would be helpful.

3) Limited Analysis Across Dataset Types:
Although evaluated on 75 datasets, the paper offers limited insight into how CLIPTTA performs under different types of distribution shift (e.g., adversarial, synthetic, or natural corruptions). Highlighting specific strengths or weaknesses across shift categories would improve understanding of the method’s robustness.

---

> ### Author Rebuttal · Authors · 2025-07-29
>
> We thank the reviewer for their insightful questions, which help us clarify our work.
>
> 1. __Q1 & W1- Runtime and memory usage:__
> To fulfill the reviewer’s request, we compare the runtime and GPU memory usage of our soft contrastive loss $L_{\text{s-cont}}$ with other gradient-based test-time adaptation methods. All methods were evaluated under the same conditions using a single A6000 GPU and a batch size of 128. As shown in the table below, CLIPTTA introduces minimal memory overhead compared to TENT and achieves competitive runtime, outperforming several recent approaches such as ETA, SAR, CLIPArTT, and WATT.  Methods like TPT [7], although designed for single-image adaptation, rely on test-time augmentations that introduce similar memory overheads and longer execution times.
>
>     | Method | Time (s)  | Memory (Gb)  |
>     |------------|------|--------------------|
>     | CLIP                 | 10.74 | 1.195 |
>     |TENT [1]           | 312 | 8.085 |
>     | ETA [2]            | 364.21| 8.252 |
>     | SAR [3]           |  704.88 | 8.085 |
>     | CLIPArTT [4]  | 334.96 | 8.087 |
>     | WATT [5]        | 1259.91 | 8.256 |
>     | TPT [7]        | 894.21  | 9.784  |
>     | $L_{\text{s-cont}}$ (ours)  | 315.01 | 8.085 |
>
>     Although $L_{\text{s-cont}}$ is a contrastive objective, **it does not require large batch sizes to perform well**. As discussed in our responses to reviewers JXSL and 26qe, CLIPTTA achieves strong performance even in the extreme case of batch size 1, thanks to the Confident Consistency Memory (CCM), which stores past confident samples. For example, on CIFAR-10, CLIPTTA reaches 93.4% accuracy with a batch size of 1, significantly outperforming TENT (40.3%) and TPT (89.8%) under the same conditions. This demonstrates that CLIPTTA remains practical for low-resource or streaming settings, where batch sizes may be limited.
>
> 2. __W2 - Unexplored Failure Cases of Contrastive Loss__:
> We agree with the reviewer that our soft contrastive loss is batch-aware and, by design, assumes the presence of sufficiently good pseudo-labels within the batch to enable effective adaptation. This reliance on pseudo-label quality is not unique to our method but shared by all gradient-based test-time adaptation (TTA) techniques, including entropy-based approaches like TENT, which can also suffer from collapse or error reinforcement under high label noise.
>
>     To empirically assess robustness in such settings, we highlight the case of ImageNet-C, where the zero-shot CLIP accuracy is only 25.1%, implying that ~75% of pseudo-labels are incorrect. Despite this, CLIPTTA reaches 41.1% accuracy, significantly outperforming all baselines, including TENT, which collapses to 17.6% (lower than CLIP's accuracy). This suggests that our batch-aware loss is considerably more robust to pseudo-label drift than existing methods, as it can still leverage confident and semantically consistent examples for adaptation, even under substantial noise.
>
>     That said, in extreme scenarios where few or no pseudo-labels are correct, adaptation may fail—as is the case for other self-supervised TTA methods. We thank the reviewer for raising this important point and will include a discussion of this limitation in the final manuscript.
>
> 3. __Q2 & W3 - Per-shift performance analysis:__
> As mentioned in the main paper (lines 63–64 and 216), we structure our evaluation around four distinct types of distribution shifts:
>     - **Synthetic corruptions**, including CIFAR-10/100-C and ImageNet-C;
>     - **Domain shifts**, comprising datasets like Sketch, Paintings, and other ImageNet variants;
>     - **Coarse-grained semantic shifts**, such as CIFAR-10 and CIFAR-100;
>     - **Fine-grained semantic shifts**, evaluated on 11 CLIP zero-shot datasets, including ImageNet.
>
>     To complement the reviewer’s request, we provide in the table below a consolidated view of CLIPTTA’s performance across these shift categories, compared to other recent TTA baselines:
>     | Method       | Corruptions | Domain Shifts | Coarse-Grained | Fine-Grained |
>     |--------------|:-----------:|:-------------:|:--------------:|:------------:|
>     | CLIP         |       40.3      |  81.3    |     78.7    |  64.7   |
>     | TENT [1]     |      35.1     |  82.3    |    83.9   |    65.3  |
>     | ETA [2]      |   42.3          |    82.3     |  84.3    |   65.4   |
>     | SAR [3]      |     48.2       |   82.1    |   82.7    |  65.3   |
>     | RoTTA [6]    |   38.7       |     80.9    |    79.0     |  62.1   |
>     | CLIPArTT [4] |  49.8      |    80.8     |    80.8    |  64.4  |
>     | WATT [5]     |  43.5       |    82.1    |     81.7     |   65.3    |
>     | TPT [7]      |    38.7      |    80.8   |    78.6     |    65.6   |
>     | TDA [8]      |   42.5      |   82.8   |     80.6    |    67.7     |
>     | CLIPTTA (ours)      |   **58.1**    |    **83.7**    |    **85.2**   |   **69.8**   |
>
>     From these results, we observe that CLIPTTA consistently outperforms all other methods across the four shift types, with particularly strong gains in challenging settings. The largest improvements occur when the base model (CLIP) performs poorly. For example, on corruption datasets, CLIP achieves 40.3% accuracy on average, while CLIPTTA reaches 58.1%—a +17.8 point gain, and 9.1 points above the second-best method.
>
>     In contrast, when CLIP already performs well—such as on domain shift datasets with 81.3% base accuracy—the gains are smaller (CLIPTTA reaches 83.7%, vs 82.8% for TDA). These results suggest that CLIPTTA’s improvement is more influenced by the base model’s initial performance than by the specific type of distribution shift. Finally, we did not identify any dataset or shift type where CLIPTTA consistently underperforms compared to other methods.
>
> 4. __Q3: Extended analysis on class collapse:__
> As requested by the reviewer, we compare the entropy of the predictions produced by CLIPTTA and several baselines over the course of adaptation on CIFAR-10-C. The table below reports the mean predictive entropy at different batch indices during test-time adaptation:
>     |  | Batch 0  | Batch 20   | Batch 40  | Batch 60 |
>     |------------|:------:|:------:|:------:|:------:|
>     | TENT [1]                                      | 2.19 | 1.75 | 1.44 | 1.19 |
>     | ETA [2]                                       | 2.17 | 2.18 | 2.18 | 2.17 |
>     | SAR [3]                                      | 2.17 | 2.13 | 2.13 | 2.15 |
>     | CLIPArTT [4]                              | 2.19 | 2.24 | 2.2 | 2.13 |
>     | WATT [5]                                     | 2.21 | 2.18 | 2.16 | 2.10 |
>     | CLIPTTA  (ours)     | __2.20__ | __2.27__ | __2.28__ | __2.29__ |
>
>     As shown, CLIPTTA maintains high predictive entropy throughout adaptation, approaching the theoretical maximum of 2.3 for CIFAR-10. This indicates effective preservation of class diversity and avoidance of class collapse, in contrast to entropy-minimization methods like TENT.
>     A similar trend is observed for the deterioration ratio: CLIPTTA maintains or improves accuracy over time, whereas methods like TENT can degrade after a few batches. However, this metric is less straightforward to interpret for approaches like ETA and SAR, which adapt to dynamically filtered subsets of the batch. We also clarify that Fig. 4b and 4c were primarily designed to compare the adaptation dynamics of our soft contrastive loss with TENT. We will consider extending the analysis to all baselines in the final version.
> 5. __Q4 - Safeguards against OOD contamination in the CCM Memory:__
> To clarify, the Class-wise Confident Memory (CCM) retains the top-k most confident samples ever observed for each class. This design ensures that only consistently high-confidence predictions are eligible for inclusion, effectively preventing the incorporation of noisy or uncertain examples. Moreover, we filter out samples detected as OOD before updating the CCM. Even if an OOD sample were to bypass this filter, it would still need to rank among the most confident examples ever seen to be included—something we did not observe in practice. Finally, the OCE loss is explicitly designed to reduce the confidence of OOD inputs, further mitigating the risk of contaminating the CCM. These safeguards together ensure the reliability of the memory.
>
> 6. __Q5: Impact of OCE loss on classification:__ We evaluate the trade-off between OOD detection and in-distribution (ID) classification accuracy when optimizing the OCE loss in Section C.4 of the supplementary material (Table 17, reproduced below). The OCE loss is weighted by a hyperparameter $\lambda_\text{oce}$.
>
>     | $\lambda_\text{oce}$ | 0 | 0.25 | 0.5 | 1 | 2 | 5 | 10 | 20 | 100 |
>     |-----|------|------|------|------|------|-----|-----|-----|-----|
>     | Acc | 67.6 | 67.6 | 67.6 | 67.6 | 67.5 | 67.3 | 66.4 | 64.5 | 56.6|
>     | AUC | 93.5 | 97.5 | 97.6 | 97.7 | 97.8 | 98.0 | 98.4 | 98.8 | 99.2 |
>     | FPR | 25.7 | 10.1 | 9.8    | 9.7   | 8.8   | 7.8   | 6.3   | 4.7   | 2.3|
>
>     As shown, the accuracy on ID samples remains remarkably stable across a wide range of $\lambda_\text{oce}$ values—from 0.25 to 5—while OOD detection improves steadily. However, when $\lambda_\text{oce} \geq 10$, ID accuracy drops despite continued gains in OOD detection, indicating a trade-off that only emerges at high values.
>
> [1] Tent: Fully test-time adaptation by entropy minimization. ICLR 2021
>
> [2] Efficient test-time model adaptation without forgetting. ICML 2022
>
> [3] Towards stable test-time adaptation in dynamic wild world. ICLR 2023
>
> [4] CLIPArTT: adaptation of CLIP to new domains at test-time. WACV 2025
>
> [5] WATT: Weight average test-time adaption of clip. NeurIPS 2024
>
> [6] Unified entropy optimization for open-set test-time adaptation. CVPR 2024
>
> [7] Test-time prompt tuning for zero-shot generalization in vision-language models.  NeurIPS 2022
>
> [8] Efficient Test-Time Adaptation of Vision-Language Models. CVPR 2024

---

> > ### Author Response · Authors · 2025-08-04
> >
> > Dear Reviewer 7qJh,
> >
> > Thank you again for your thoughtful and constructive review.
> >
> > We hope that our rebuttal has addressed your concerns, in particular the questions related to the efficiency of CLIPTTA, its robustness in challenging conditions, the performance across different types of distribution shifts, and the safeguards designed to ensure the reliability of the memory module.
> >
> > We would be glad to further clarify any remaining questions or points of confusion you might have.

---

> > > ### Comment · Reviewer_7qJh · 2025-08-05
> > >
> > > Thank you for the detailed rebuttal. I appreciate the authors’ clarifications, which have resolved my major concerns. In light of the additional explanations and new results, I will raise my score to accept.
> > >
> > > That said, I would strongly recommend incorporating the newly provided results into the final version of the paper. Additionally, I believe the discussion of the CCM component in the methods section could be expanded for better clarity and completeness. It would also strengthen the paper to include a more explicit discussion of its limitations, which is currently lacking.
> > >
> > > Overall, I appreciate the authors' efforts in addressing the concerns, and I look forward to seeing these improvements reflected in the final version.

---

### Comment · Area_Chair_yEsm · 2025-08-05
**Discussion with authors**

Dear Reviewers 4cbr,  JXSL,

Please go through the author rebuttal and clarifications and let the authors know if you have any remaining concerns, so that they are able to respond timely. If the response is satisfactory, mention that too.

Also, please complete the Mandatory Acknowledgement only after the discussion.

Thanks,
AC

---

### Note · Authors · 2025-08-14

We thank the reviewers for their constructive feedback, which has helped us strengthen the paper. The main concerns have been addressed through additional experiments, clarifications, and planned updates for the revised version.
- **Generality and comparisons to related work**: We extended experiments to additional CLIP backbones (ResNet50, ViT-B/32, ViT-L/14) and SigLIP, showing consistent gains. We also expanded comparisons to include RPL and BATCLIP and found that CLIPTTA outperforms both across all tested scenarios. These new results will be added to the supplementary material and in Table 1 in Section 4.1, respectively.

- **Adaptation loss alignment**: Our batch-aware soft contrastive loss was primarily motivated by its alignment with CLIP’s pre-training objective. We also observed apparent robustness to class collapse compared to entropy minimization. Both factors likely contribute to its effectiveness, though disentangling them is challenging. We will present this motivation more cautiously in the revised version, as suggested by the reviewers.

- **CCM and single-sample setting**: We clarified that our Class-wise Confident Memory (CCM) enables strong adaptation even at batch size 1, as discussed in the supplementary material. This table will be moved to the main paper, and the CCM explanation in Section 3 will be expanded for clarity, as suggested by the reviewers.

- **Additional ablations**: Additional ablations, including the cross-dataset benchmark and class collapse analysis, confirm our theoretical analysis that the soft contrastive loss is inherently more robust to class collapse than entropy minimization. These results will be added to Section 4.2.

- **Clarifications**: We will clearly state that the logit scale parameter is not updated during adaptation in Section 3.1 and make the ID/OOD definitions in Section 3.3 more explicit.

We sincerely appreciate the reviewers’ constructive feedback and openness during the discussion period, which has greatly improved the clarity, scope, and completeness of this work. Two reviewers have indicated that they intend to raise their scores to 4 and 5, which we believe reflects the strengthened contribution.

---

### Decision · Program_Chairs · 2025-09-17

**Decision:**

Accept (poster)

**Comment:**

This paper proposes a test-time adaptation framework for vision-language models like CLIP, termed CLIPTTA. They show the effectiveness of the method for both closed and open-set settings. The reviewers appreciated the clear writing, problem addressed (ID and OOD) and extensive evaluation. Based on the overall contributions and the positive reviews, the recommendation is to accept the paper. The authors are strongly encouraged to address the remaining concerns regarding explaining the limitations of the work, etc. in the final version.